# Regulatory T cells in the mouse hypothalamus control immune activation and ameliorate metabolic impairments in high-calorie environments

Maike Becker [1,2], Stefanie Kälin[2,3], Anne H. Neubig [1,2], Michael Lauber[1,2], Daria Opaleva [1,2], Hannah Hipp[1,2], Victoria K. Salb[1,2], Verena B. Ott[2,3], Beata Legutko[3], Roland E. Kälin[4,5,6], Markus Hippich [2,7], Martin G. Scherm[1,2], Lucas F. R. Nascimento[1,2], Isabelle Serr[1,2], Fabian Hosp[8], Alexei Nikolaev[9], Alma Mohebiany [9], Martin Krueger[10], Bianca Flachmeyer[10], Michael W. Pfaffl [11], Bettina Haase [12], Chun-Xia Yi [13], Sarah Dietzen[14], Tobias Bopp [14], Stephen C. Woods[15], Ari Waisman [9], Benno Weigmann [16], Matthias Mann [8], Matthias H. Tschöp [2,3] ✉ & Carolin Daniel [1,2,17] ✉

The hypothalamus in the central nervous system (CNS) has important functions in controlling systemic metabolism. A calorie-rich diet triggers CNS immune activation, impairing metabolic control and promoting obesity and Type 2 Diabetes (T2D), but the mechanisms driving hypothalamic immune activation remain unclear. Here we identify regulatory T cells (Tregs) as key modulators of hypothalamic immune responses. In mice, calorie-rich environments activate hypothalamic CD4⁺ T cells, infiltrating macrophages and microglia while reducing hypothalamic Tregs. mRNA profiling of hypothalamic CD4⁺ T cells reveals a Th1-like activation state, with increased *Tbx21*, *Cxcr3* and *Cd226* but decreased *Ccr7* and *S1pr1*. Importantly, results from Treg loss-of function and gain-of-function experiments show that Tregs limit hypothalamic immune activation and reverse metabolic impairments induced by hyper-caloric feeding. Our findings thus help refine the current model of Treg-centered immune-metabolic crosstalk in the brain and may contribute to the development of precision immune modulation for obesity and diabetes.

Metabolic diseases such as obesity and Type 2 diabetes (T2D) are among the most severe health threats of modern society. To control systemic metabolism, the central nervous system (CNS) receives neuronal and endocrine inputs reflecting the current state of metabolism[1–3]. Specialized brain areas in several hypothalamic regions, including the arcuate nucleus (ARC)[4], integrate this feedback to maintain a balanced metabolic profile and a stable body weight[1,2].

From an immune perspective, immune activation in the CNS that is initiated as a consequence of exposure to calorie-rich diets has been suggested to promote impaired metabolic control, thereby contributing to the development of obesity and T2D[5]. In the CNS, the ARC is particularly affected by high-fat, high-sugar (HFHS) diets that cause impairments of its microenvironment[6–10]. In addition, the adjacent circumventricular organ, the median eminence, contains a highly fenestrated vascular endothelium that represents an incomplete

blood-brain barrier. Thus, together with the ARC, the median eminence has been considered as an important 'window' for communication between the mediobasal hypothalamus and the circulation. More recently, another avenue of crosstalk between the brain and circulation has been described that is consistent with the early description by Rennels et al. of a physical interconnection between the cerebrospinal fluid (CSF) and the perivascular spaces around the brain vasculature[11]. Whereas classical lymphatic vessels in the CNS meninges permit drainage of large particles and immune cells from the brain to cervical lymph nodes[12–14], the so-called 'glymphatic system' allows interstitial solutes to be cleared from the brain via exchange of CSF and interstitial fluid[15,16]. Moreover, vascular channels were found to link the bone marrow of the skull to the brain surface[17].

High-calorie environments were shown to promote chronic low-grade inflammation in metabolic tissues, while tissue damage emerges as a result of dysregulated chronic innate and adaptive immune responses[18]. Recent literature suggested that a reduction in fat-residing regulatory CD4[+]T cells (Tregs) upon high-calorie challenges sustained local inflammation and worsened metabolic parameters[19,20]. However, it remains currently unknown whether Tregs residing in brain regions relevant for metabolic control do control local immune activation and do contribute to the regulation of systemic metabolism.

In the brain, as innate immune sensors of pathological changes, microglia are the primary resident immune cell and maintain a functionally optimal local environment[21–23].

Despite these insights, the mechanisms permitting functional surveillance of the CNS during steady-state conditions remain incompletely understood. It has been suggested that systemic T lymphocytes including CD4[+] T cells transit the CNS and participate in immune surveillance[24–26]. Consistent with this, earlier work indicated that under physiological conditions immune cells in the perivascular and subarachnoid spaces can provide immune surveillance of the CNS[12,13,27,28]. In addition, recent work by Rustenhoven et al. underscored the importance of the dural sinuses as a critical neuroimmune interface, where T cells encounter antigens to support immune surveillance. Specifically, this work showed efflux of CSF to the dural sinuses, permitting accumulation of brain-enriched antigens[29].

From an adaptive immune perspective, in addition to their classical contribution in autoimmune settings such as multiple sclerosis, the last years have seen a growing appreciation of the role of brain-residing CD4[+] T cells in behavior, neurodegenerative and neuroinflammatory diseases[30–35] and most recently also in the healthy brain[29,34,36–38]. Work from Adrian Liston and his team characterized the CD4[+] T-cell population in the healthy brain, indicating in situ initiation of a residency program[36].

In spite of these findings, the specific adaptive immune cells, which are relevant for controlling immune responsiveness and which are compromised in the brain by environmental challenges such as calorie-rich diets are unknown. Foxp3[+]CD4[+] Tregs are strong candidates for controlling local immune reactivity. They can be induced de novo[39,40], maintain immune homeostasis[41] and can control tissue integrity, inflammation and function, for example in visceral white fat[19,20] as well as in the CNS[42,43]. More importantly, we and others demonstrated that Tregs respond to environmental-metabolic signaling cues e.g., environmental temperature and diet composition in non-lymphoid tissues and thereby control tissue function and inflammation[44,45].

Here, we expose mice to a high-caloric diet challenge and determine the transcriptome of hypothalamus-residing CD4[+]T cells to identify the receptors relevant for recruitment of T cells to and retention at inflammatory sites, i.e., the hypothalamus. Using Treg-specific gain- and loss-of-function studies we further demonstrate that Foxp3[+]Tregs regulate hypothalamic immune responsiveness. Specifically, hypothalamus-residing Foxp3[+]Tregs are significantly reduced upon exposure to a HFHS diet. Furthermore, Treg transfers in vivo significantly reduce hypothalamic innate immune activation of mice fed a calorie-rich diet and Treg gain-of-function studies improve systemic metabolic parameters. Therefore, our results identify a key function of Foxp3[+]Tregs in shaping innate immune responsiveness in response to hypercaloric environments in the hypothalamus of the CNS.

## Results

### Identification of CD4[+]T cells and CD25[+]Foxp3GFP[+]Tregs in the brains of healthy mice

First, to define and confirm the presence of CD4[+]T cells in the steady state from mice fed a standard diet, we used immunofluorescence to stain against CD4 and GFAP in mouse brains (Fig. 1a). To identify regulatory Foxp3[+]Tregs we likewise employed immunofluorescence with staining against CD3 and Foxp3 (Fig. 1b). As second approach in an independent laboratory, we used diaminobenzidine (DAB) stainings against GFP in Foxp3GFP reporter mice as a mean to confirm the identification of Tregs in healthy brains with light microscopy upon transcardial perfusion (Fig. 1c). These immunohistochemical analyses demonstrated Foxp3[+]GFP[+]Tregs within the parenchyma of young lean Balb/c Foxp3GFP reporter mice fed a standard diet (Fig. 1c).

We identified a substantial proportion of T cells in the brain in close proximity to vessels, as assessed by immunofluorescence and co-staining against CD4 and the vessel marker lectin (Fig. 1d–f). However, and in line with recent observations[36–38], a small number of T cells was found to be located distal to the vasculature (Fig. 1d–f). These findings are in line with a non-vascular non-meningeal placement for a certain subset of T cells[36]. Moreover, recent studies have highlighted that vessel-associated T cells, which can be localized in the perivascular space, can secrete soluble mediators to facilitate crosstalk with the neighboring parenchyma such as neurons, astrocytes and microglia, the latter well known for their critical role in maintaining and supporting neurological functionality[31,32,34].

As an independent approach, we employed high-dimensional flow cytometry to identify and quantify CD4[+]T cells including Treg cells in the brain of mice. Given the critical role of the hypothalamic region in directing metabolic control, we focused on T cells that are located in the hypothalamus vs. the remaining brain (rest brain from hereon) for flow cytometry analyses. Using a set of exclusion markers (CD8a, CD11b, CD11c, CD14, F4/80, B220) and a dead-cell stain (sytox or eF450), we identified local CD4[+]T cells (Fig. 1g) including CD4[+]CD25[+]Foxp3GFP[+]Tregs (Fig. 1g, h) in hypothalami of healthy mice. CD4[+]T cells, including Foxp3[+]Tregs, were likewise observed in other parts of the brain (Fig. 1g, h). In line with recent observations by Pasciuto et al., transcardial perfusion did neither abolish the CD4[+]T cells in the rest brain or hypothalamus of healthy mice (Supplementary Fig. 1a), nor were Treg frequencies reduced (Supplementary Fig. 1b). Furthermore, upon transcardial perfusion, T cells from these healthy mice presented with an activated phenotype as assessed by CD62L[low] staining (Supplementary Fig. 1c).

### A high-calorie diet promotes hypothalamic immune activation in local CD4[+]T cells

Next, to study immuno-metabolic alterations in local brain-residing CD4[+]T cells, mice were exposed for various time frames to a HFHS diet composed of 58.0% kcal from fat, 25.5% kcal from carbohydrates and 16.4% kcal from proteins for 8–48 weeks (wk). To assess whether hypercaloric challenges might impinge on the co-localization of T cells with vessel markers, we used immunofluorescence to stain against CD4 and the vessel marker lectin in mouse brains (Fig. 2a, b). Similar to standard-diet conditions, upon exposure to a hypercaloric challenge, the majority of CD4[+]T cells identified in mouse brains remained vessel-associated (Fig. 2c). No differences in total counts of CD4[+]T cells that were vessel-associated vs. distinct from vessel were observed upon exposure to standard vs. HFHS diet (Fig. 2d, e). To validate these findings in an alternate approach, we performed DAB stainings. To this

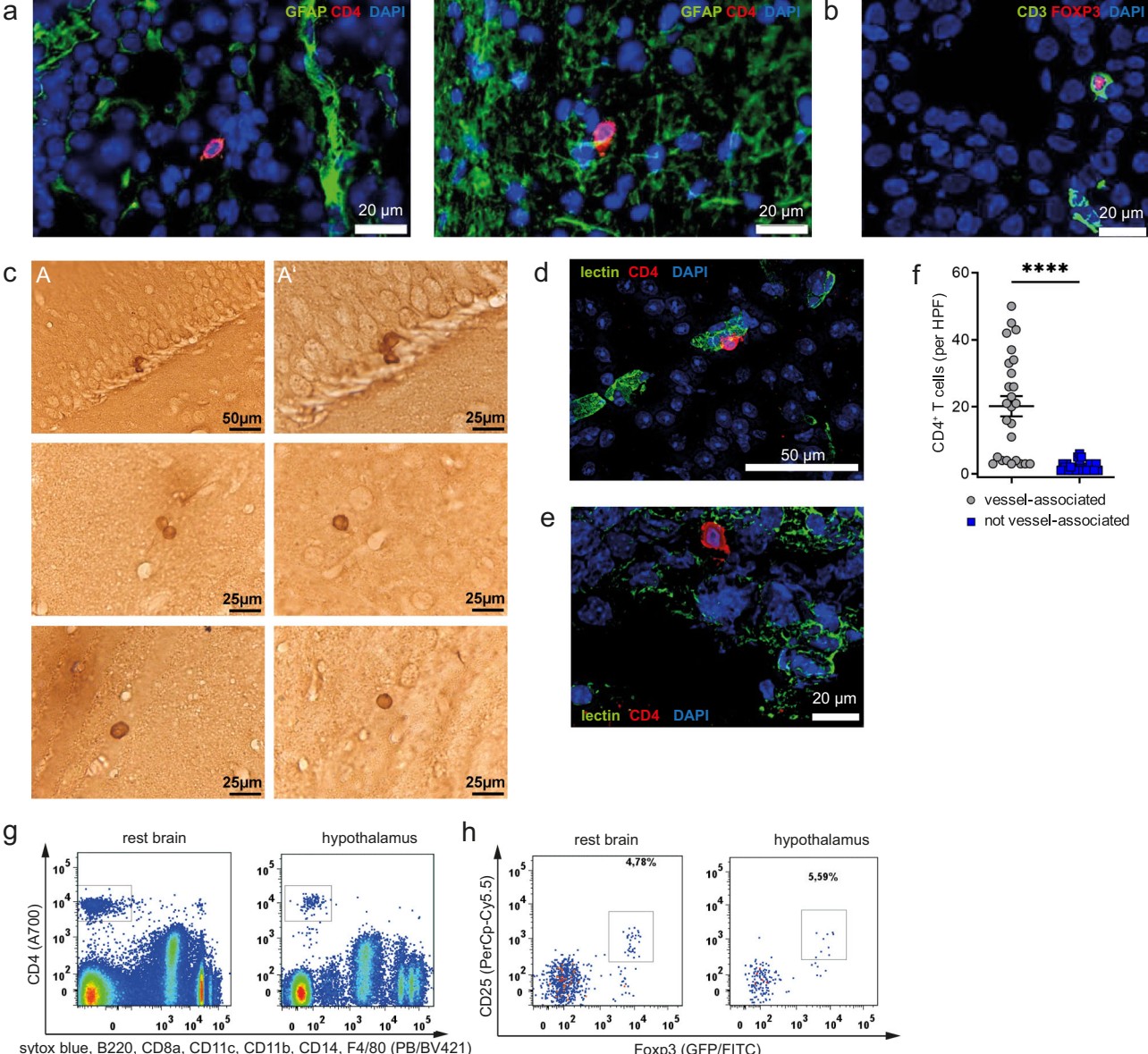

**Fig. 1 | Identification of CD4⁺T cells and CD4⁺CD25⁺Foxp3GFP⁺Tregs in brains of healthy lean mice. a, b** Representative immune fluorescence staining of brain cryosections (12 μm) of Balb/c mice (**a**) stained for glial fibrillary acidic protein (GFAP, astrocyte marker, green), CD4 (T-cell marker, red) and DAPI (nuclei, blue) and **b** stained for CD3 (T-cell marker, green), Foxp3 (Treg marker, red) and DAPI (nuclei, blue). Stainings were obtained from >3 independent experiments. The scale bar is 20 μm. **c** Anti-GFP DAB staining of 30 μm vibratome sections of brains from Foxp3GFP Balb/c reporter mice to detect Tregs in the brain. The scale bar is 50 μm or 25 μm as indicated. Stainings were obtained from >3 independent experiments. **d, e** Representative immune fluorescence staining of brain cryosections (12 μm) stained for lectin (vessel, green), CD4 (CD4 T-cell marker, red) and DAPI (nuclei, blue) for the quantification of vessel-associated (**d**) vs. vessel distinct (**e**). The scale

bar is 50 μm or 20 μm as indicated in the figure. **f** Summary graph for the quantification of vessel-associated (depicted as circles) vs. distinct from vessel (depicted as squares) CD4⁺T cells. Dots represent counts per high power field (HPF) as in (**d**, **e**), obtained from multiple sections. Mean ± SEM. Two-tailed unpaired Student's *t* test, *P* < 0.0001. **g, h** Flow cytometric analysis of **g** CD4⁺T cells from mouse brain without hypothalamus (referred to as rest brain) and hypothalamus. CD4⁺T cells are analyzed as being positive for CD4 and negative for a panel of exclusion markers (live dead stain, B220, CD8a, CD11b, CD11c, CD14, F4/80). **h** Flow cytometric analysis of CD25⁺Foxp3GFP⁺ Tregs (% of live CD4⁺) from mouse rest brain and hypothalamus. Tregs were pre-gated on CD4⁺ and negative for exclusion markers as in (**g**). Source data are provided as a Source Data file. ****P < 0.0001.

end, we obtained CD4GFP mice by crossing mice expressing the Cre recombinase under the *Cd4* promoter with mice containing a *loxP*-flanked STOP sequence followed by the eGFP gene inserted into the Rosa26 locus. This enables the identification of T cells by GFP expression. We employed antibodies directed against GFP in brains from such CD4GFP mice exposed to standard vs. HFHS diet referring to a lean vs. an obese state respectively (Fig. 2f). The observed low T-cell frequencies impeded a robust quantification using light microscopy.

Therefore, to permit an in-depth characterization and quantification of hypothalamus-residing CD4⁺T cells in response to hypercaloric challenges, we focused on high-dimensional flow cytometry analyses. Canonical neuroinflammatory diseases, many of which are induced by reactions to CNS-derived self-antigens and pathogens, do involve high numbers of infiltrating CD4⁺T cells. In contrast to these classical neuro-immunological disease settings, only a very mild trend towards an increase in local CD4⁺T-cell frequency occurred after chronic exposure to a HFHS diet (Supplementary Fig. 1d). At the same

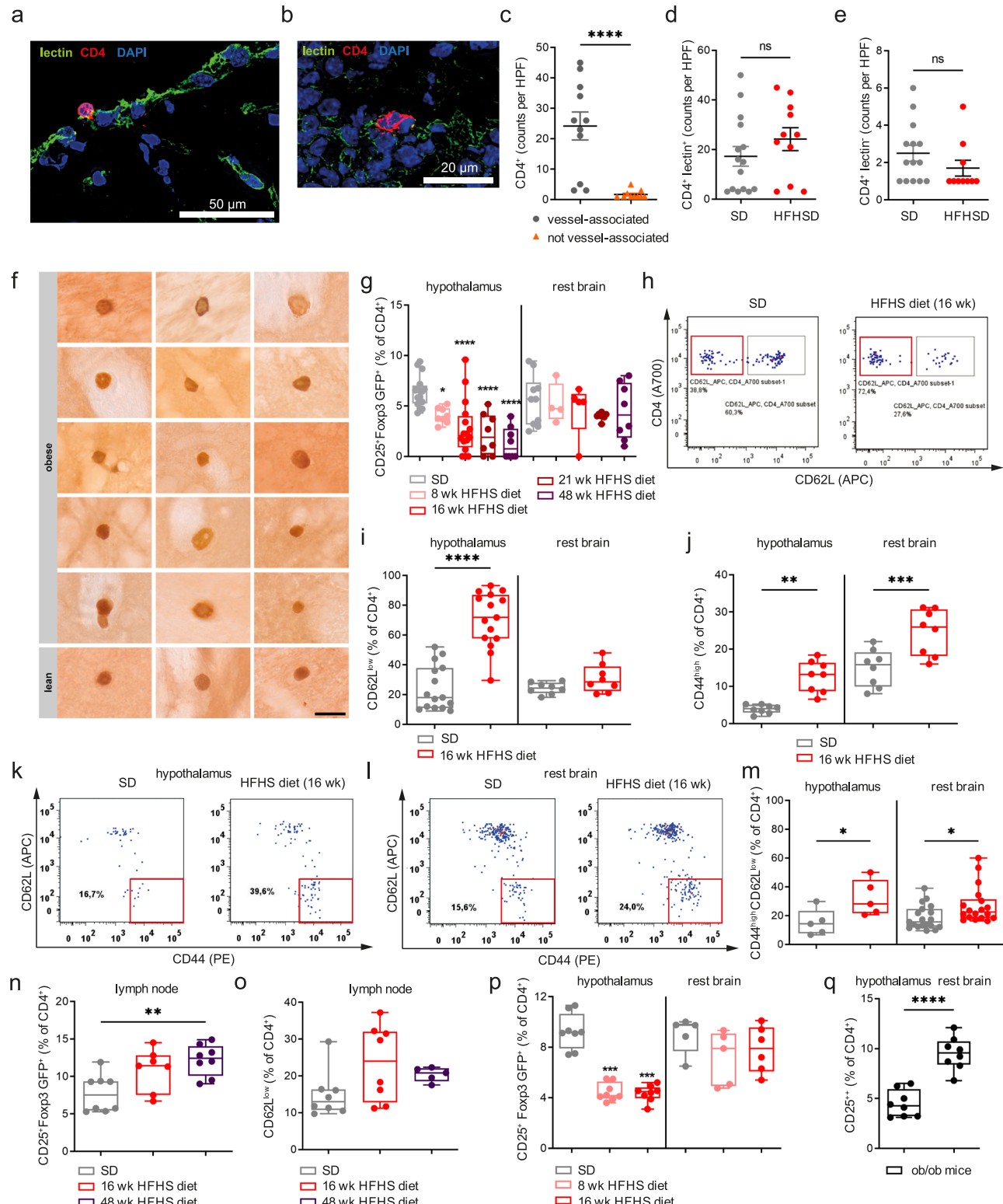

time, we observed a significant and progressing decline of hypothalamic CD4+CD25+Foxp3GFP+Tregs (Fig. 2g). The reduction of hypothalamus-residing Foxp3+Tregs was also present after a short-term exposure to HFHS diet (8 weeks), a time when the Balb/c mice were not yet obese (Supplementary Fig. 1e). These results point to a potentially important influence of diet composition in modulating the hypothalamic immuno-metabolic microenvironment.

Supporting the notion of local immune activation triggered by hypercaloric challenge, mice fed a high-caloric diet displayed an enhanced proportion of hypothalamus-residing CD4+T cells with an activated phenotype (Fig. 2h, i). Likewise, activated CD44hiCD4+T cells distinctly increased in the hypothalamus as well as in the remaining parts of the brain (Fig. 2j). Activation of CD4+T cells upon HFHS diets was confirmed by increased frequencies of CD4+CD62LlowCD44highT cells in the hypothalamus (Fig. 2k, m) and to a lesser extent also in the remaining parts of the brain (Fig. 2l, m).

The observed decline in hypothalamic Treg cells together with the increase in local CD4+T-cell activation did not simply reflect immune

**Fig. 2 | A high-calorie diet promotes hypothalamic immune activation in local CD4⁺T cells.** **a, b** Representative immune fluorescence staining of Balb/c brain cryosections (12 μm) stained for lectin (vessel, green), CD4 (CD4 T-cell marker, red) and DAPI (nuclei, blue) for the quantification of vessel-associated (**a**) vs. vessel distinct (**b**) after 16 weeks HFHS diet. The scale bar is 50 μm or 20 μm as indicated. **c** Summary graph for the quantification of vessel-associated vs. not vessel-associated CD4⁺T cells upon exposure to high-fat high-sugar (HFHS) diet for 16 weeks. Vessel-associated CD4⁺ T cells are depicted as circles, T cells distinct from vessels as triangles and were quantified as counts per high power field (HPF) across multiple sections. Mean ± SEM. Two-tailed Mann−Whitney U test, P < 0.0001. **d, e** Summary graph for the quantification of **d** vessel-associated CD4⁺T cells and **e** T cells distinct from vessels per high power field (HPF) upon exposure to a standard diet (SD) or high-fat high-sugar (HFHS) diet for 16 weeks. Mean ± SEM. Two-tailed Mann−Whitney U test **d** P = 0.1531; **e** P = 0.2198. (**f**) Representative anti-GFP DAB immunostainings in brains of CD4GFP mice fed a standard diet or HFHS diet for 6 months. Vibratome sections were cut with 30 μm and the scale bar is 20 μm. Stainings were obtained from >3 experiments. **g** Quantification of CD25^hiFoxp3⁺Treg frequencies (% of CD4⁺) in rest brain vs. hypothalamus of Foxp3GFP Balb/c mice exposed to HFHS diet for 8–48 weeks. n = 4−16 biological replicates from ≥4 independent experiments. Depicted are box-and-whisker plots (min to max with all data points). One-way ANOVA with Šidák post hoc test, multiple comparisons vs. lean values. Adjusted P values for hypothalamus: P(8 weeks) = 0.0660; P(16 weeks) <0.0001; P(21 weeks) <0.0001; P(48 weeks) <0.0001. Adjusted P values for rest brain: P(8 weeks) > 0.9999; P(16 weeks) = 0.9882; P(21 weeks) = 0.6420; P(48 weeks) = 0.9131. **h** Representative FACS plots for the analysis of activated CD62L^low CD4⁺ T cells in mouse Foxp3GFP Balb/c brains with or without exposure to HFHS diet (SD gray, 16 weeks HFHSD red). **i** Quantification of (H) in % of CD4⁺ T cells. Depicted are box-and-whisker plots (min-to-max values with all data points); n = 8 or n = 15 biological replicates from >4 independent experiments. One-

way ANOVA with Šidák post hoc test. Adjusted P values: P(hypothalamus) <0.0001; P(rest brain) = 0.6028). **j** Box-and-whisker plots (min-to-max values with all data points) for activated CD44^hiCD4⁺ T cells from mouse Foxp3GFP Balb/c brains exposed to SD or HFHS diet; n = 8 biological replicates from four independent experiments. One-way ANOVA with Šidák post hoc test, adjusted P values: p(hypothalamus) = 0.0011; P(rest brain) = 0.0004). **k–m** Representative FACS plots for hypothalamic CD44^hiCD62L^low effector memory T cells (as % of CD4⁺) from the hypothalamus (**k**) or rest brain (**l**) of Foxp3GFP Balb/c mice. **m** summary plot depicted as box-and-whisker plots (min to max with all data points). N = 5−19 biological replicates. One-way ANOVA with Šidák post hoc test, adjusted P values: p(hypothalamus) = 0.0323; P(rest brain) = 0.0381). **n, o** Lymph node-residing CD4⁺Foxp3GFP⁺Tregs and CD4⁺CD62L^low T cells of Foxp3GFP Balb/c mice fed the standard or the HFHS diet. n = 8 biological replicates from 2 to 4 independent experiments. Data are depicted as box and whisker plot (min-to-max values with all data points). One-way ANOVA with Šidák post hoc test, adjusted P values: P(SD vs. 16 weeks) = 0.0699; P(SD vs. 48 weeks) = 0.0056; P(16 weeks vs. 48 weeks) = 0.6880). **o** One-way ANOVA with Šidák post hoc test, adjusted P values: P(SD vs. 16 weeks) = 0.1038; P(SD vs. 48 weeks) = 0.5281; P(16 weeks vs. 48 weeks) = 0.8587). **p** Quantification of Treg frequencies from rest brain vs. hypothalamus of Foxp3GFP C57Bl/6J mice exposed to standard diet or HFHS diet for 8 or 16 weeks. Depicted are box-and-whisker plots (min to max with all data points). n = 4−8 biological replicates from 2 to 4 independent experiments. One-way ANOVA with Šidák post hoc test, adjusted P values: hypothalamus: P(SD vs. 8 weeks) <0.0001; P(SD vs 16 weeks) <0.0001; P(8 vs. 16 weeks) >0.9999; rest brain: P(SD vs. 8 weeks) = 0.2470; P(SD vs. 16 weeks) = 0.6984; P(8 vs. 16 weeks) = 0.9663). **q** Treg frequencies in the hypothalamus and rest brain of genetically obese ob/ob mice on SD. Depicted are box-and-whisker plots (min to max with all data points). n = 8 biological replicates from two independent experiments. Two-tailed unpaired t test, P < 0.0001. Source data are provided as a Source Data file. *P < 0.05; **P < 0.01; ***P < 0.001; ****P < 0.0001.

activation occurring in the peripheral immune system upon exposure to the HFHS diet. Specifically, these changes in Treg frequencies were not reflected in peripheral lymph nodes (Fig. 2n). Long-term exposure to such hypercaloric challenges promoted a significant increase of Tregs in LNs (Fig. 2n), which could result among others from inflammatory expansion or retention. Furthermore, we did not identify significant changes in the activation of CD4⁺T cells (Fig. 2o) purified from mesenteric LNs of mice subjected to the HFHS diet, albeit we observed a mild but significant increase in IL-17A production of such CD4⁺T cells (Supplementary Fig. 2a). Given the limited numbers of Tregs within the hypothalamus, it is not feasible experimentally to directly assess their cellular functionality through Treg suppression assays. However, we have performed such Treg suppression assays using Tregs isolated from LNs of either mice exposed to HFHS diet or to standard diet. These functional studies did not reveal any significant impairments in Treg function per se in mice exposed to the HFHS diet (Supplementary Fig. 1j). Nevertheless, our data indicate that the deteriorating effects of hypercaloric feeding on Tregs is a local effect as observed specifically in the hypothalamus.

C57Bl/6J Foxp3GFP reporter mice, which are more prone to diet-induced obesity than Balb/c mice (Supplementary Fig. 1e, f), likewise had significantly reduced frequencies of hypothalamus-residing Foxp3⁺GFP⁺Tregs when maintained on the HFHS diet (Fig. 2p). To understand to what extent the altered CD4⁺T cell and Treg profile upon HFHS diet promotes inflammatory downstream processes that may also radiate to other regions of the brain, we assessed classical inflammatory mediators associated with NFκB. Specifically, animals exposed to HFHS diet showed significantly higher gene expression levels of Ccl5 induced by NFκB signaling in the hypothalamus (Supplementary Fig. 2b). Trends towards higher expression of Ccl5, cRel and RelB were also observed in the remaining brain (Supplementary Fig. 2c). Involvement of NFκB signaling upon HFHS diet exposure was confirmed when we assessed isolated cellular populations. Specifically, ACSA2⁺ astrocytes purified from the hypothalamus of HFHS diet-exposed mice showed a significant increase in Ikkb expression when compared to standard diet-fed animals (Supplementary Fig. 2d). No

significant changes in Ikkb and related Tnfa expression in astrocytes were seen in remaining parts of the brain (Supplementary Fig. 2d). Purified microglia showed trends towards increased Ikkb and Tnfa expression in both the hypothalamus and remaining brain (Supplementary Fig. 2d). Likewise, similar trends for Ikkb and Tnfa abundance were seen in NeuN⁺ neurons in the remaining brain (Supplementary Fig. 2d). Given the importance of neuronal-immune crosstalk upon hypercaloric challenge, we validated the expression of NFκB-related signaling in purified NeuN⁺ neurons in an independent setting of HFHS-diet-exposed Balb/c mice. Specifically, in NeuN⁺ neurons from animals fed HFHS diet we observed a significantly increased abundance of Ccl5, cRel, Relb, Ikkb and Tnfa (Supplementary Fig. 2e).

In an alternate experimental setting, we confirmed the decline in Tregs in the hypothalamus by analyses of genetically obese ob/ob mice which lack expression of the hormone leptin. Likewise, those obese mice showed a reduced frequency of hypothalamic Tregs when compared to the rest of the brain (Fig. 2q). Exposure of ob/ob mice to HFHS diets also promoted a decline in Tregs in remaining parts of the brain (Supplementary Fig. 2f, g).

## Transcriptome analysis of hypothalamus-residing CD4⁺T cells confirms diet-induced immune activation

The control of the properties of brain/hypothalamus-residing CD4⁺T cells in particular in response to HFHS diet is currently unknown. Thus, in a pilot RNA-seq experiment, we performed transcriptome analyses of hypothalamus-residing CD4⁺T cells. Specifically, we focused on CD4⁺CD25⁻CD44^low+intermediate T cells with a rather lower activation profile to permit the assessment of priming effects induced by hypercaloric immuno-metabolic challenges[46]. To this end, mice were fed standard or HFHS diet, CD4⁺CD25⁻CD44^low+int T cells and CD4⁺CD25⁺Foxp3GFP⁺Tregs were cell-sorted and T-cell populations were pooled from groups of n = 9 animals (Supplementary Fig. 3a). The sort-purified T-cell populations from standard diet vs HFHS diet-fed animals did not differ with respect to CD44 MFI on a cellular level (Supplementary Fig. 3b). In parallel experiments, we performed RNA-seq of sort-purified hypothalamus-residing CD4⁺CD25⁻ and CD4⁺CD25⁺⁺T cells from groups of n = 7 ob/ob

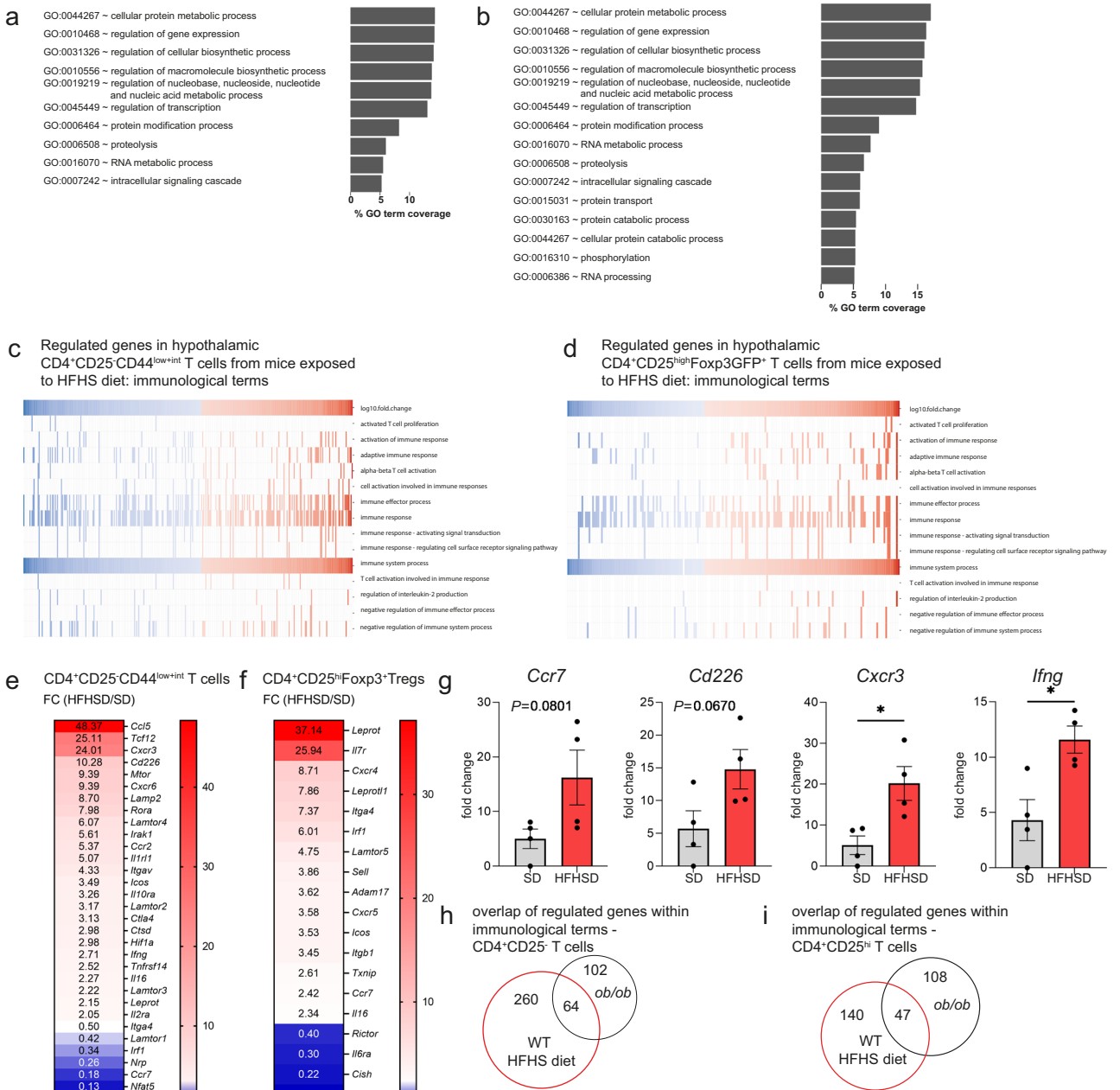

**Fig. 3 | Transcriptome of hypothalamus-residing CD4+T cells reflects diet-induced immune activation. a–d** DESeq2-normalized read counts regulated more than 2.5-fold (HFHS diet vs. SD) were functionally annotated to Gene Ontology Biological Processes (GOBP) level 5 using DAVID Bioinformatics Resources 6.7. Terms are depicted as percentage GO-term gene coverage for **a** hypothalamic CD4+CD25−CD44low+int T cells and **b** hypothalamic CD4+CD25highFoxp3GFP+T cells from Foxp3GFP Balb/c mice exposed to 16 weeks SD or HFHS diet (pooled from n = 9 animals). Regulated genes were annotated to selected immunological GOBP terms and color-coded by log10-fold change; **c** hypothalamic CD4+CD25−CD44low+int T cells and **d** CD4+CD25highFoxp3GFP+T cells from Foxp3GFP Balb/c mice exposed to HFHS diet. **e, f** Heatmap of the fold change of selected differentially regulated

genes in response to HFHS diet vs. SD in hypothalamic CD4+CD25−CD44low+int T cells (**e**) or CD4+CD25highFoxp3GFP+T cells (**f**) of Foxp3GFP Balb/c mice. **g** Gene expression analysis of FACS-sorted CD4+CD25−CD44low+int T cells from Foxp3GFP Balb/c mice exposed to 16 weeks SD or HFHS diet by qPCR. n = 4 biological replicates per group from two independent experiments. Mean ± SEM. Unpaired two-tailed t test, P(Ccr7) = 0.0801; P(Cd226) = 0.0670; P(Cxcr3) = 0.0188; P(Ifng) = 0.0169. **h, i** VENN diagram of regulated genes within selected immunological GO terms of hypothalamic (**h**) CD4+CD25− T cells and **i** CD4+CD25high T cells from *ob/ob* mice or C57Bl/6J wild-type mice fed the HFHS diet. Source data are provided as a Source Data file. *P < 0.05.

and WT mice. A general overview of GO-term gene coverage in hypothalamic CD4+CD25−CD44low+int T cells (Fig. 3a), as well as in CD4+CD25+Foxp3GFP+Tregs (Fig. 3b), highlights regulation at cellular protein metabolic processes, regulation of gene expression and regulation of cellular biosynthetic processes. Similar overviews of GO-term gene coverage in hypothalamic CD4+CD25−T-cell populations from *ob/ob* mice fed a standard diet are depicted in Supplementary

Fig. 3c, d. In addition, an overview of top-regulated genes in both CD4+CD25−Foxp3GFP−CD44low+int T cells and CD4+CD25+Foxp3GFP+Tregs from mice exposed to HFHS diet or from standard diet-fed *ob/ob* mice is outlined in Supplementary Fig. 3g–j.

To dissect the observed T-cell activation seen after high-caloric challenge in cellular analyses at a molecular level, we next focused on pathway analyses related to immune effector responses and T-cell

activation (Fig. 3c, d). This revealed that when the mice were exposed to the HFHS diet, hypothalamus-residing CD4+T cells had distinctly regulated genes involved in cellular migration to non-lymphoid tissues, and genes involved in the recruitment of T cells to as well as their retention at inflamed tissue sites (Fig. 3e, f). Specifically, analysis of hypothalamic CD4+CD25−Foxp3GFP−CD44low+int T cells from mice with hypercaloric feeding revealed upregulation of *Cxcr3*, NF-kB target gene *Ccl5* and *Ccr4* accompanied by a downregulation in *Ccr7* and *S1pr1* expression. A reduced expression of *S1pr1* is critically involved in regulating tissue retention of T cells[47,48], while downregulation of *Ccr7* promotes their migration to inflamed tissues and the ability to have immediate effector functions.

## A high-calorie diet promotes a Th1-like immune activation in hypothalamic CD4+T cells

Enhanced activation of locally-residing CD4+T cells was confirmed after HFHS feeding; e.g., by increased expression of *Pi3kγ*, which drives priming and activation of T cells, and which is important for chemotaxis, migration and promotion of inflammation[49]. After exposure to the HFHS diet, CD4+T cells also had enhanced expression of *Mtor* and *Map3k7* (encoding Tak1), which likewise induce activation and differentiation of CD4+T cells[50] (Fig. 3c, e). Moreover, maintenance on the HFHS diet resulted in upregulation of *Tlr4* expression in hypothalamic CD4+T cells, thereby promoting immune activation. The expression of *Tlr4* has been linked to CNS inflammation[51] although until now reports have largely focused on innate immune cells.

The immune activation profile of hypothalamic CD4+T cells of mice exposed to the HFHS diet revealed Th1 characteristics as evidenced by upregulation of *Tbx21* (T-bet), *Cxcr3* and *Cd226*, a profile which has been found to be specifically expressed in Th1 inflammatory T cells[52] (Fig. 3c, e). The comparison of gene expression profiles in hypothalamic CD4+T cells from *ob/ob* mice revealed an overlap in regulation, and likewise pointed toward a Th1-mediated inflammatory setting indicated by upregulation of *Cxcr6, Ifng, Cd226* and *Irak1* (VENN diagram of regulated genes within immunological terms in Fig. 3h, VENN diagram of all regulated genes in Supplementary Fig. 3e, overview in Supplementary Fig. 5c).

To confirm the results of the RNA-seq experiment, we analyzed the gene expression of FACS-sorted CD4+CD25−Foxp3GFP−CD44low+int T cells from mouse brains exposed to standard vs HFHS diet in an independent sample set by quantitative real-time qPCR (Fig. 3g). *Ccr7* and *Cd226* expression showed a trend towards an increase upon exposure to HFHS diet, while *Cxcr3* and *Ifng* expression were significantly increased and thus, confirmed the Th1 polarization of CD4+T cells isolated from mouse brains upon hypercaloric challenge. In another sequencing experiment, we compared FACS-sorted activated conventional T cells (gated as CD4+CD25−Foxp3GFP−CD44highCD62Llow) from brains and hypothalami of standard diet vs HFHS diet mice (Supplementary Fig. 4a–g) to see, if the hypercaloric challenge induces phenotypic differences in the activated conventional CD4+ T-cell subset. Here, gene set enrichment analyses (GSEA) revealed that this very defined T-cell subset undergoes proliferation (gene sets related to cell cycle were enriched, see Supplementary Fig. 4h, i) in response to HFHS diet. The expansion of this T-cell subset is in line with our cellular analyses performed by flow cytometry (Fig. 2m). Phenotypically, we did not identify major significant changes in response to hypercaloric challenge (Supplementary Fig. 4a–d, h).

## Transcriptome of hypothalamic CD4+CD25+Foxp3+Tregs reveals diet-induced immune activation

CD4+CD25+Foxp3GFP+Tregs purified from the hypothalami of mice exposed to the HFHS diet had upregulated migratory characteristics and T-cell recruitment such as *Adam10* (Fig. 3d). In addition, there was increased expression of *Rock1* and *Irak1* in Treg populations, genes which in turn mediate activation and promote inflammatory profiles. *Irf1* expression was also upregulated, indicating repressed Foxp3

expression[53]. High levels of *Mtor* expression in Tregs reflect their high-glycolytic activity and a short-lived effector Treg phenotype[54]. Furthermore, upregulation of *Tnfrsf14* expression[55] in Tregs from HFHS diet-exposed animals might support impaired Treg suppressive function (Fig. 3e). In addition, the second most highly upregulated transcript in CD4+CD25+Foxp3GFP+Tregs from HFHS-fed mice is *Il7r*. Accordingly, previous work showed that memory Tregs do require IL-7 for their maintenance in the skin[56]. Therefore, this concept could suggest that upon exposure to HFHS diet the upregulation of *Il7r* could constitute rather a compensatory mechanism to safeguard their maintenance during hypercaloric challenge. On the contrary, recent findings revealed that the upregulation of IL-7Rα expression strongly interfered with IL2 receptor signaling in Treg cells by competing for the common γ-chain, accompanied by blunted downstream Stat5 phosphorylation[57]. These results could indicate a reduced proliferative potential of Tregs in response to HFHS diet based on high levels of *Il7r* expression due to reduced IL2 receptor signaling. (Overlap in regulation in CD4+CD25hiT cells from *ob/ob* mice is outlined in Fig. 3i and in Supplementary Fig. 3d).

## Foxp3+Tregs reduce diet-induced hypothalamic immune activation of macrophages and microglia

To dissect the functional role of Foxp3+Tregs in regulating diet-induced hypothalamic immune activation we first focused on alterations in immune responsiveness of macrophages and microglia. Specifically, CD45hiCD11b+ cells are commonly addressed as macrophages of the perivascular and subarachnoid compartments[23,58,59], whereas CD45intCD11b+-expressing cells are considered as microglia, which originate from yolk sac-derived macrophage progenitors and populate the brain parenchyma during prenatal development[60,61]. We found that short-term exposure to the HFHS diet (1 week) promotes a distinct increase in infiltrating macrophages (Fig. 4a, b), thereby amplifying innate immune responsiveness.

Moreover, under physiological conditions, microglia have low expression of the major histocompatibility complex class II (MHCII)[62]. The mean fluorescent intensity (MFI) for MHCII of microglia purified from the brains of mice exposed to the HFHS diet significantly increased in response to high-caloric challenge (Fig. 4e), reflecting an increase in MHCII molecules present on the surface per cell. After exposure to the HFHS diet, flow cytometric analyses revealed an increase in MHCII expression to intermediate levels. These MHCII levels were significantly higher when compared to those from mice exposed to a standard diet (Supplementary Fig. 6a–e). Similarly, the MFI of costimulatory markers such as CD80/86 and CD40 was significantly enhanced by the high-caloric challenge (Fig. 4c–f). Likewise, frequencies of CD80high- and CD86high-expressing microglia increased upon high-caloric feeding (Supplementary Fig. 6f, g). This increased innate immune responsiveness in response to HFHS diets was confirmed by analyses which demonstrated enhanced numbers of MHCII/CD80 and MHCII/CD86 double-positive microglia (Supplementary Fig. 6h, i) thereby underlining their potential role as functional antigen-presenting cells.

To validate MHCII expression within microglial cells we next used Cx3cr1GFP reporter mice. While Cx3cr1 expression alone is not sufficient to distinguish the resident microglial population from infiltrating mononuclear phagocytes or perivascular, meningeal or choroid plexus macrophages[63–65], the combination with flow cytometric analysis of intermediate levels of CD45 and CD11b+ refines the microglia population. Accordingly, we characterized CD45intCD11bCx3cr1GFP+ cells from the brains of Cx3cr1GFP reporter mice subjected to 12 weeks of HFHS diet or maintained on standard diet. Exposure to the HFHS diet distinctly enhanced the percentages of CD45intCD11b+Cx3cr1GFP+MHCIIhigh cells when compared to animals on the standard diet (Fig. 4g).

Moreover, immunofluorescence analyses of brains for MHCII and Iba1 not only showed the well described increase in Iba1 staining

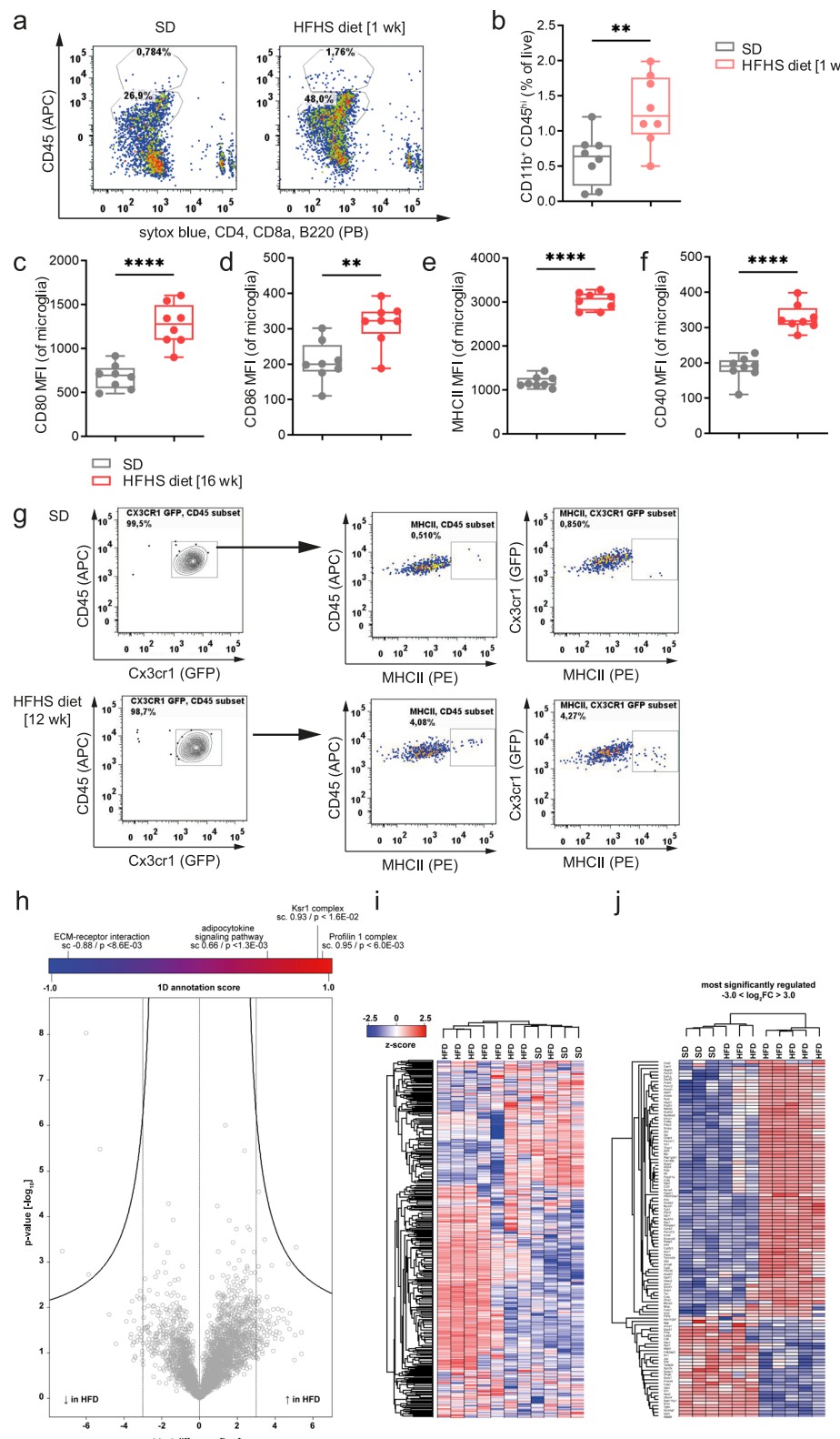

intensity in mice exposed to a HFHS diet, but confirmed levels of microglial MHCII expression, especially within the choroid plexus, median eminence and tissue near to meninges (Supplementary Fig. 6k).

Next, to further link diet-induced innate immune activation of glial cells with local T cells, we employed quantitative, high-resolution mass spectrometry (MS)-based proteomics[66]. We purified microglia from animals exposed to various durations of HFHS diet to dissect their phenotypes and immune activation. Specifically, we isolated CD45[int]CD11b[+]Cx3cr1GFP[+] from brains of Cx3cr1[GFP] reporter mice subjected to 36 weeks of HFHS or standard diet. First, we pairwise compared the proteomes between standard and HFHS-diet microglia (Fig. 4h and post-sort purity in Supplementary Fig. 6j). In silico 1D annotation enrichment of the quantitative proteome differences[67]

**Fig. 4 | A high-calorie diet promotes immune activation of microglia and infiltrating macrophages. a** Representative FACS plots for CD45$^{int}$CD11b$^+$ microglia and CD45$^{hi}$CD11b$^+$ infiltrating macrophages in brains from Foxp3GFP Balb/c reporter mice fed the standard or the HFHS diet. **b** Frequencies of CD45$^{hi}$CD11b$^+$ macrophages after 1 week of HFHS diet. Depicted are Box-and-whisker plots (min-to-max values with single data points), $n = 8$ from two independent experiments. Two-tailed unpaired $t$ test; $P = 0.0064$. **c–f** Analyses of MHCII expression and costimulatory markers (CD80, CD86 and CD40) on CD45$^{int}$CD11b$^+$ microglia of Foxp3GFP Balb/c mice fed a standard or HFHS diet for 16 weeks. Depicted are Box-and-whisker plots (min-to- max values with single data points), $n = 8$ biological replicates from two independent experiments. Two-tailed unpaired $t$ test; $P$(CD80) < 0.0001; $P$(CD86) = 0.0028; $P$(MHCII) < 0.0001; $P$(CD45) < 0.0001. **g** Representative FACS plots for MHCII expression on hypothalamic microglia of

Cx3cr1$^{GFP}$ reporter mice exposed to standard or HFHS diet for 12 weeks. **h–j** Volcano plot (**h**) of the pairwise comparison between proteomes of CD45$^{int}$CD11b$^+$Cx3cr1GFP$^+$ microglia fed a standard diet or exposed to the HFHS diet for 36 weeks. Expression fold changes (t-test difference, log$_2$) were calculated and plotted against the $t$ test $P$ value (−log$_{10}$). 1D annotation enrichment for the most enriched or depleted annotation term (Corum, GOBP, GOCC, GOMF, KEGG, Pfam) is depicted on top of the volcano plot including the Benjamini−Hochberg-corrected false-discovery rate. **i** All proteins quantified in the total proteome analysis were grouped using unsupervised hierarchical clustering of the z-scored MaxLFQ-intensities across the indicated experimental groups. **j** Most significantly regulated proteins of **i** were grouped using unsupervised hierarchical clustering of the z-scored MaxLFQ-intensities across the indicated experimental groups. Source data are provided as a Source Data file. **$P$ < 0.01; ****$P$ < 0.0001.

identified the adipocytokine signaling pathway (involving proteins including Pck2, Prkag1, Acsl1, Nfkb1, Rela, Mtor, Prkab1 and Stat3) as one of the most enriched annotation terms (Corum, GOBP, GOCC, GOMF, KEGG, Pfam, Fig. 4h) in microglia of mice exposed to the HFHS diet ($P < 1.3 \times 10^{-3}$). Next, we grouped all proteins quantified in the total proteome analysis using unsupervised hierarchical clustering of the z-scored MaxLFQ-intensities across the indicated groups (microglia HFHS diet vs. microglia standard, Fig. 4i). Upon exposure to the HFHS diet, the phenotype of microglia indicated immune activation (upregulation of innate immune responses, $P = 6.47 \times 10^{-28}$; e.g., Hspd1, Rps6ka3, Mapkapk2, Mapkapk3, Sin3a and Pik3ap1). Moreover, pathway analyses for HFHS-diet microglia revealed upregulation of proteins related to glial cell differentiation ($P = 1.44 \times 10^{-8}$; including Cnp, Nfia or Gap43), glial cell development ($P = 1.44 \times 10^{-8}$; e.g., Itgam and Plp1), type-2 diabetes ($P = 1.5 \times 10^{-16}$; e.g., Mapk1, Hk1, Hk2, Pkm, Mapk3, Mtor and Pik3cd), the insulin-signaling pathway ($P = 3.22 \times 10^{-36}$; e.g., Mapk1, Hk1, Fasn, Ppp1cb, Hk2, Kras, Hras) and Alzheimer's disease ($P = 1.41 \times 10^{-58}$; e.g., Ndufs2, Ndufs1, Mapk1, Uqcr10, Atp5o, Atp5e, Ndufa5, Atp5f1 and Atp5a1). In addition, the most prominently regulated proteins as outlined in Fig. 4h were grouped using unsupervised hierarchical clustering (Fig. 4j). Thus, having received a hypercaloric challenge followed by diet-induced obesity, microglia enter a reactive state of immune activation, thereby promoting local immune activation that might also impact neighboring cells such as neurons.

## Loss of Treg function exacerbates diet-induced immune activation in the hypothalamus

Next, to underline the contribution of Foxp3$^+$Tregs in regulating hypothalamic immune responsiveness we used a set of loss-of-function experiments. First, Tregs were depleted using established procedures involving anti-CD25 depleting antibodies[44,68] (the experimental setup is outlined in Supplementary Fig. 7a). mCD25 antibodies applied $i.p.$ resulted in successful peripheral and hypothalamic Treg depletion (Supplementary Fig. 7b, c). Importantly, peripheral Treg depletion was sufficient to promote a distinct immune activation in hypothalamus-residing microglia (Fig. 5a–f) and infiltrating macrophages of both C57Bl/6J and Balb/c mice (Fig. 5g–i). Specifically, 1 week after Treg depletion, microglia from hypothalami of mice fed a standard diet had increased costimulatory markers (CD80 and CD86, C57Bl/6J mice, Fig. 5b) and enhanced antigen-presenting potential as evidenced by higher MHCII expression when compared to control mice (Fig. 5c).

In a second set of loss-of-function experiments, we used Foxp3 DTR+ vs Foxp3 DTR- mice[69] and Tregs were depleted upon administration of diphtheria toxin in Foxp3 DTR+ mice (DTX, $i.p.$) (Fig. 6, scheme in Supplementary Fig. 7d; Exemplary stainings upon successful depletion of peripheral and hypothalamic Tregs are presented in Supplementary Fig. 7e, f). Next, in Treg-depleted Foxp3 DTR+ mice, 1-week exposure to the HFHS diet promoted a significantly stronger immune activation as indicated by frequencies and phenotypes of

infiltrating macrophages (Fig. 6a, b), the activation status of microglia and macrophages (Fig. 6c–j).

To further dissect the role of Foxp3$^+$Tregs in impacting microglial immune responsiveness and function we performed proteomic analyses of microglia purified from Foxp3 DTR mice with or without Treg depletion based on DTX application and a short challenge of HFHS diet. Accordingly, we isolated CD45$^{int}$CD11b$^+$ microglia from the brains of Foxp3 DTR mice 1 week after Treg depletion. First, we pairwise compared the proteomes between microglia from Treg-depleted (Foxp3 DTR+) and Treg competent (Foxp3 DTR-) animals exposed to the HFHS diet for 2 weeks (Fig. 6k). In silico 1D annotation enrichment of the quantitative proteome differences identified a mild upregulation of pathways involved in activation of immune responses, antigen processing and presentation, innate and inflammatory immune effector processes, positive activation of NFκB transcription factor activity, Alzheimer's disease and Parkinson's disease. On an individual protein level, we found the A kinase-anchoring molecule 13 (Akap13) to be most prominently enriched in microglia from Treg-depleted animals ($P = 0.0301$). Akap13 has been reported to be involved in the regulation of TLR2-mediated microglia activation, and Akap13 was also suggested to function as an incipient Alzheimer's disease-related gene[70].

In line with these findings, following Treg depletion and exposure to the HFHS diet, mice also had enhanced local T-cell activation in other parts of the brain. Moreover, Treg depletion in Foxp3 DTR+ mice exposed to the high-caloric challenge resulted in an increase in hypothalamus-residing activated CD44$^{hi}$CD4$^+$T cells (Fig. 6l) that was accompanied by enhanced CD4$^+$T-cell proliferation in the hypothalamus (Fig. 6m).

## Gain of Treg function in the hypothalamus reduces immune activation and improves metabolic impairments upon hypercaloric feeding

Next, to mechanistically dissect the functional role and ability of Foxp3$^+$Tregs in regulating diet-induced hypothalamic immune activation and to define the resulting impact on systemic metabolism, we performed a series of in vivo gain-of-function experiments. First, we employed specific Treg expansion with anti-IL2/IL2 antibody complexes[71] followed by exposure to HFHS diet. Specifically, 6 µg anti-IL2/IL2 antibody complexes were injected $i.p.$ on 3 consecutive days in C57Bl/6J mice, and the HFHS diet was started at the peak of Treg expansion at day 5 (Fig. 7, the experimental scheme is provided in Supplementary Fig. 8a). The expansion was maintained by $i.p.$ injections twice a week. Control mice received vehicle followed by HFHS diet application. Of note, local Treg expansion resulted in a significant decline in HFHS-diet-induced *Ccl5* expression in the hypothalamus (Fig. 7d), highlighting the anti-inflammatory features of local Tregs. Similarly, *Ccl5* and related NFκB targets showed trends toward a reduction in remaining parts of the brain of mice receiving specific Treg expansion together with reductions in NFκB-related genes as *RelB*, *cRel* and *Tnfa* (Supplementary Fig. 8c). Importantly, to link reduced local inflammation upon Treg expansion with systemic

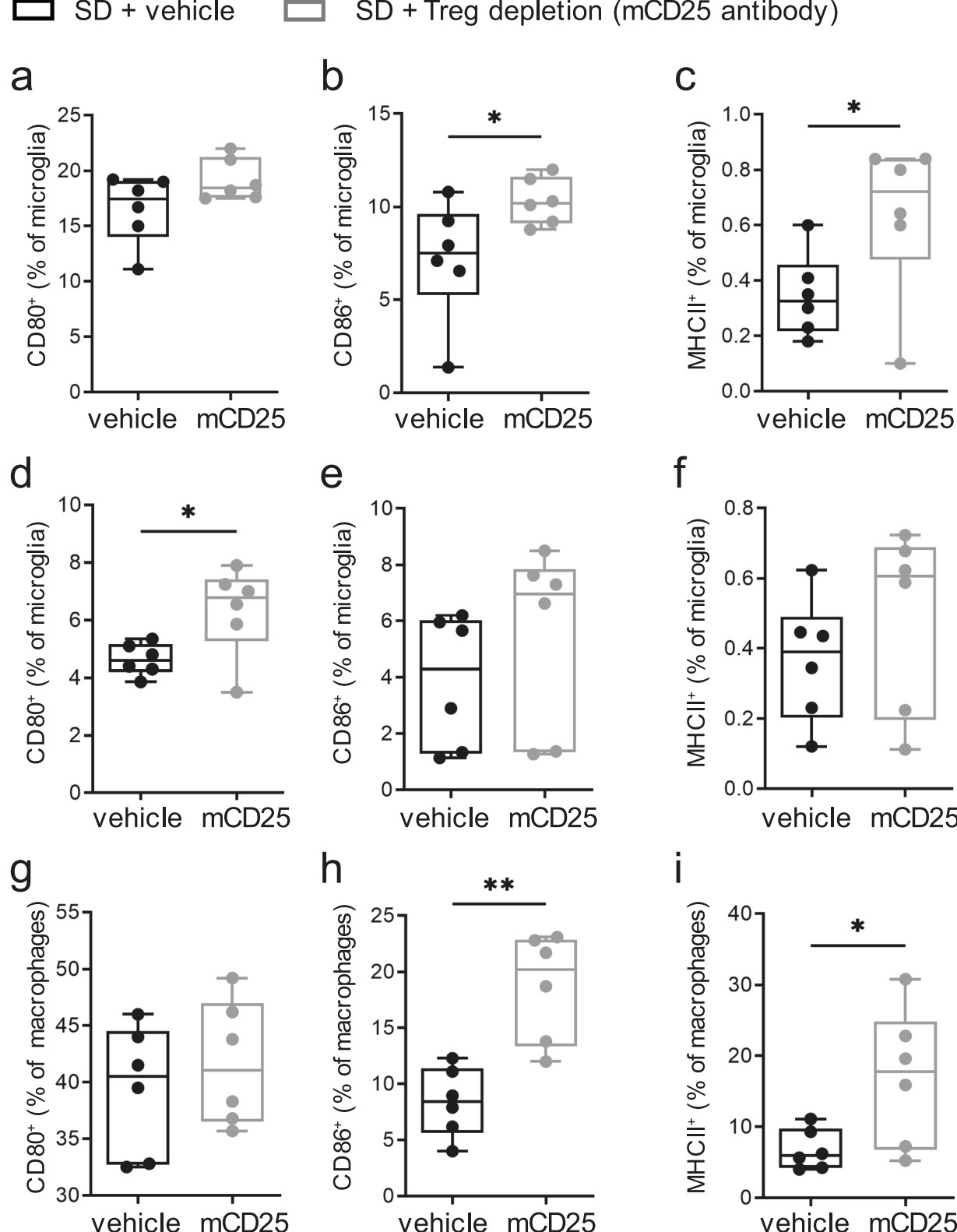

**Fig. 5 | Treg loss-of-function experiments reflect their impact in lowering diet-induced hypothalamic immune activation. a–f** Analyses of frequencies of CD80+, CD86+, and MHCII+ cells in CD45intCD11b+ microglia of **a–c** C57Bl/6J or **d–f** Balb/c mice on standard diet examined 2 weeks after Treg depletion using mCD25 antibodies. Depicted are Box-and-whisker plots (min-to-max values with single data points). $n = 6$ biological replicates from two independent experiments. Two-tailed unpaired $t$ test, $P(\mathbf{a}) = 0.1054$; $P(\mathbf{b}) = 0.0499$; $P(\mathbf{c}) = 0.0491$; $P(\mathbf{d}) = 0.0292$; $P(\mathbf{e}) = 0.3581$; $P(\mathbf{f}) = 0.3513$. **g–i** Identification of frequencies of CD80+, CD86+, and MHCII+ cells in CD45hiCD11b+ macrophages of C57Bl/6J mice on standard diet examined 2 weeks after Treg depletion using mCD25 antibodies. Depicted are Box-and-whisker plots (min-to-max values with single data points). Two-tailed unpaired $t$ test, $P(\mathbf{g}) = 0.4956$; $P(\mathbf{h}) = 0.0013$; $P(\mathbf{i}) = 0.0326$. Source data are provided as a Source Data file. *$P < 0.05$. **$P < 0.01$.

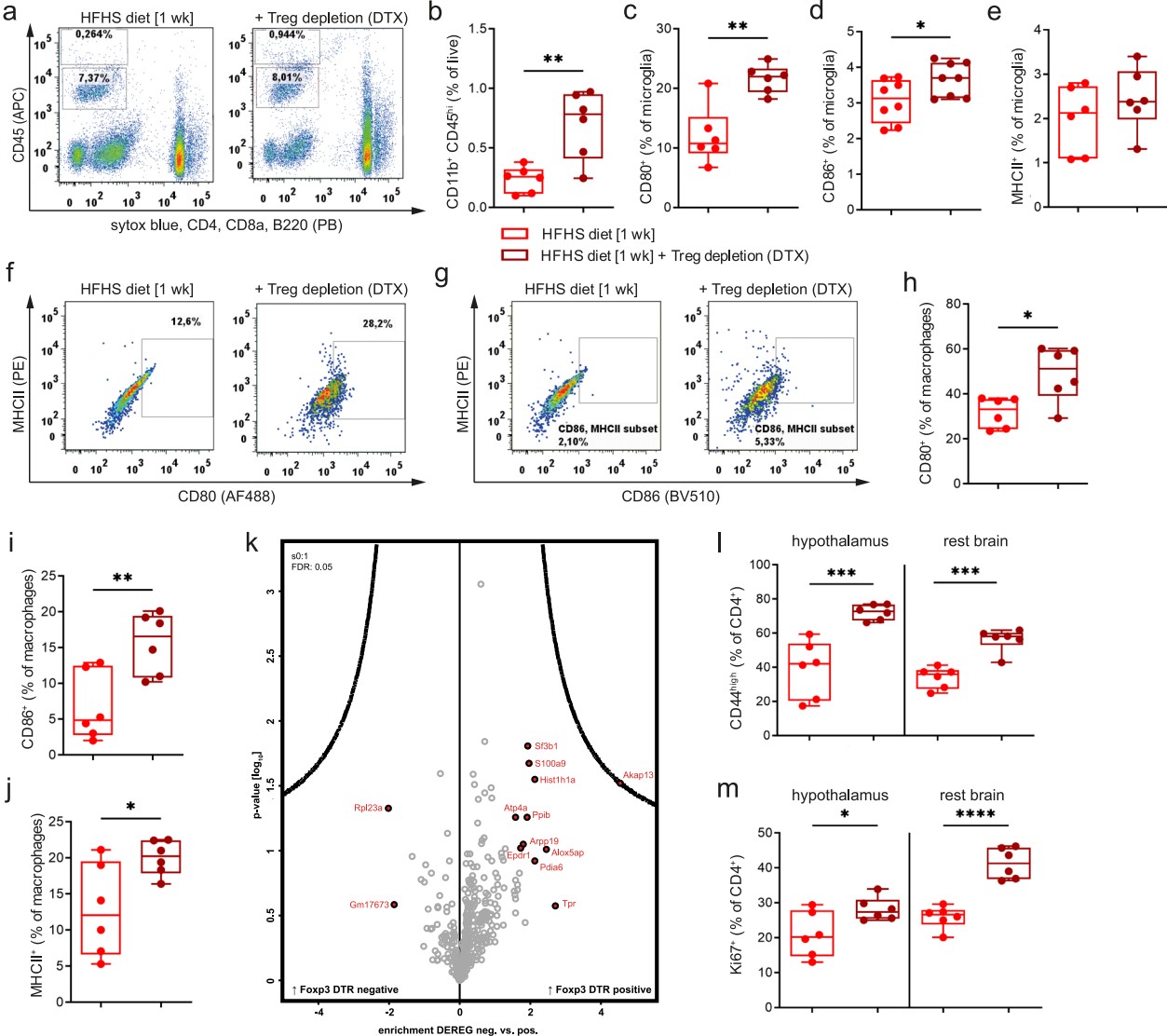

**Fig. 6 | Loss of Foxp3⁺Tregs enhances diet-induced hypothalamic immune activation.** Tregs were depleted using *i.p.* administration of diphtheria toxin (DTX) in Foxp3 DTR mice. **a** Representative FACS plots for macrophages and microglia after maintenance on the HFHS diet for 1 week with or without Treg depletion using Foxp3 DTR mice and ablation with DTX. **b** Quantification of CD45$^{hi}$CD11b$^{+}$ infiltrating macrophages. Data are depicted as box-and-whisker plots (min-to-max values with single data points). $N = 6$ biological replicates. Two-tailed unpaired Student's $t$ test with $P = 0.0040$. **c–e** Immune activation of CD45$^{int}$CD11b$^{+}$ microglia as seen from CD80$^{+}$, CD86$^{+}$ and MHCII$^{+}$ expressing cells. Data are depicted as box-and-whisker plots (min-to-max values with single data points). $N = 6$–8 biological replicates. **f–j** Representative FACS plots and quantification of immune activation in infiltrating macrophages as seen from CD80$^{+}$, CD86$^{+}$ and MHCII$^{+}$ expressing cells.

Data are depicted as box-and-whisker plots (min-to-max values with single data points). $N = 6$ biological replicates. **k** Volcano plot of proteome analysis of FACS-sorted CD45$^{int}$CD11b$^{+}$ microglia from Foxp3 DTR mice without ($n = 4$) or with ($n = 6$) DTX-mediated Treg depletion exposed to 2 weeks HFHS diet. **l, m** Activation status and proliferative capacity of brain-residing CD4$^{+}$T cells (1 week HFHS diet) with or without DTX-mediated Treg depletion. $n = 4$–6 biological replicates per group from two independent experiments. Depicted are Box-and-whisker plots (min-to-max values with single data points). Two-tailed unpaired $t$ test, $P$(**b**) = 0.0040; $P$(**c**) = 0.0014; $P$(**d**) = 0.0440; $P$(**e**) = 0.3135; $P$(**h**) = 0.0124; $P$(**i**) = 0.065; $P$(**j**) = 0.0272; $P$(**l**, hypothalamus) = 0.0008; $P$(**l**, rest brain) = 0.0001; $P$(**m**, hypothalamus) = 0.0346; $P$(**m**, rest brain) <0.0001; Source data are provided as a Source Data file. *$P < 0.05$; **$P < 0.01$; ****$P < 0.0001$.

metabolic readouts, we assessed body weight and fat mass. Specific Treg expansion fostered a significant reduction in body weight and fat mass gain in response to HFHS diet (Fig. 7c, e). Next, we performed a series of *i.p.* glucose tolerance tests to further define the impact of Treg expansion on systemic energy metabolism. Here, Treg gain-of-function supported a significantly improved glucose tolerance following 4 vs. 8 weeks of HFHS diet (Fig. 7f, g).

In a second experimental setup, we employed established procedures using cold acclimation to foster Tregs[44,45] (scheme in Supplementary Fig. 8f). One week of cold exposure resulted in a significant increase of hypothalamus-residing Foxp3⁺Tregs (Supplementary Fig. 8g). In the hypothalamus, costimulatory molecules CD80 and

CD86 on macrophages were significantly declined in response to cold-induced Treg enhancements (Supplementary Fig. 8i, j), while MHCII expression on macrophages remained unaffected (Supplementary Fig. 8k). Likewise, cold-mediated Treg induction led to a significant decline in CD86 expression on microglia (Supplementary Fig. 8m), while CD80 and MHCII only showed a trend towards a reduction (Supplementary Fig. 8l and S8n).

To validate the relevance of Treg gain-of-function in a third independent experimental setting, we used a direct cellular approach and performed peripheral transfers (*i.v.*) of Foxp3⁺GFP⁺Tregs into C57Bl/6J-recipient mice followed by exposure to the HFHS diet (for scheme see Supplementary Fig. 9a, b). Analyses of hypothalami 2

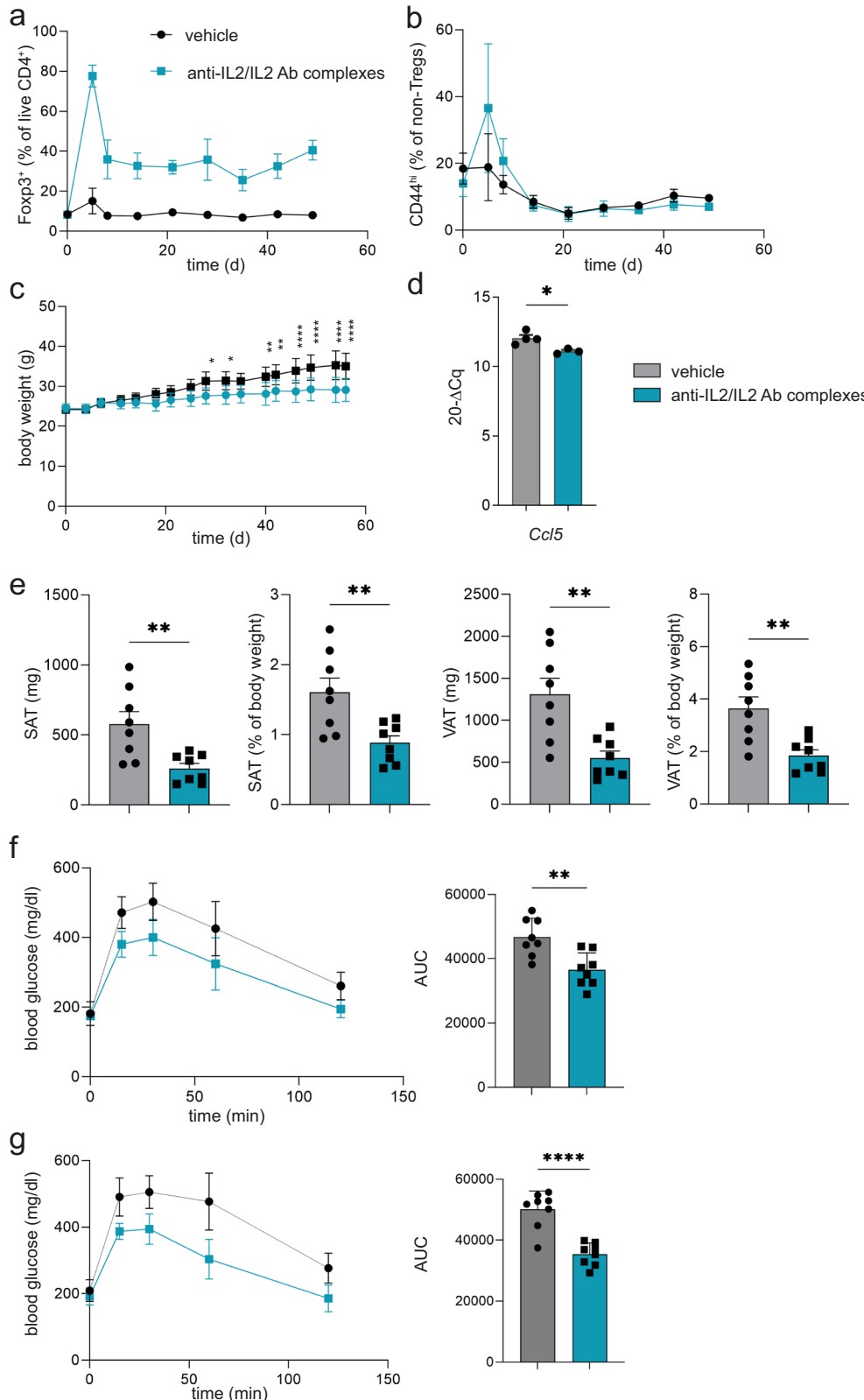

**Fig. 7 | In vivo Treg expansion by anti-IL2/IL2 antibody complexes improves metabolic indices in mice fed a HFHS diet. a**, **b** Treg frequencies (**a**) and activation of non-Tregs (**b**) analyzed by flow cytometry in peripheral blood upon Treg expansion using anti-IL2/IL2 antibody complexes. $N = 9$ biological replicates per group. Mean ± SD. Two-way RM ANOVA with Šidák post hoc test, exact $P$ values are provided in the Source Data file. **c** Corresponding body weight curves from (**a**). $n = 9$ per group. Mean ± SD. Two-way RM ANOVA with Šidák post hoc test. Exact $P$ values are provided in the Source Data file. **d** *Ccl5* gene expression analysis of hypothalami from (**a**).

Mean ± SEM. Two-tailed unpaired $t$ test, $P = 0.0193$. **e** Subcutaneous adipose tissue (SAT) and visceral adipose tissue (VAT) weights at the end of the study represented in grams and % of body weight. Each dot represents a biological replication. Mean ± SEM. Two-tailed unpaired $t$ test, $P$(SAT, mg) = 0.0053; $P$(SAT, %BW) = 0.0064; $P$(VAT, mg) = 0.0027; $P$(VAT, %BW) = 0.0026. **f**, **g** ipGTT after 4 weeks (**f**) and 8 weeks (**g**) of Treg expansion. Each dot represents a biological replication. Mean ± SD. Two-tailed unpaired $t$ test, $P$(4 weeks) <0.0001; $P$(8 weeks) <0.0001. Source data are provided as a Source Data file. *$P < 0.05$; **$P < 0.01$; ***$P < 0.001$; ****$P < 0.0001$.

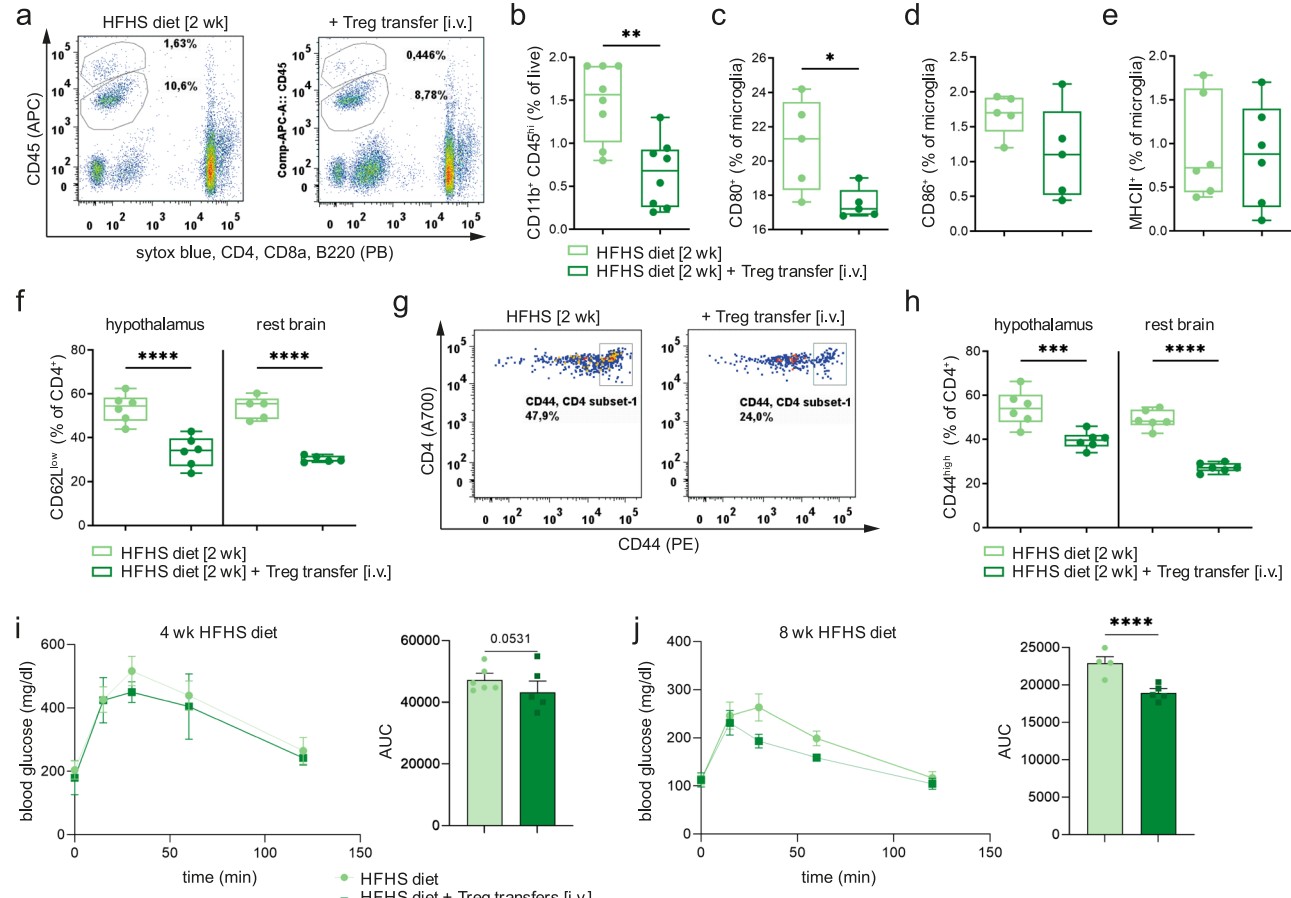

**Fig. 8 | Transferred Foxp3+Tregs reduce hypothalamic immune activation induced by the HFHS diet and improve metabolic indices. a** Representative FACS plots for CD45intCD11b+ microglia and CD45hiCD11b+ infiltrating macrophages in brains from C57Bl/6J mice fed the HFHS diet for 2 weeks with or without peripheral Foxp3GFP+Treg transfer (*i.v.*). **b** Box-and-whisker plots (min-to-max values with single data points), for the frequencies of infiltrating macrophages of mice. *n* = 8 biological replicates from two independent experiments. Two-tailed unpaired *t* test, *P* = 0.0053. **c–e** Identification of frequencies of CD80+ (*P* = 0.0256), CD86+ (*P* = 0.1207) and MHCII+ (*P* = 0.8314) cells in CD45intCD11b+ microglia. Box-and-whisker plots (min-to-max values with single data points). *n* = 6 biological replicates from two independent experiments. Two-tailed unpaired *t* test. **f** Frequencies of brain-residing CD4+CD62LlowT cells. *n* = 6 biological replicates per group from two

independent experiments. Box-and-whisker plots (min-to-max values with single data points). Two-tailed unpaired *t* test, *p*(hypothalamus) = 0.0004; *P*(rest brain) <0.0001. **g, h** Representative FACS plots (**g**) for the identification of activated CD4+CD44hiT cells in mice with or without *i.v.* Treg transfer. **h** Quantification of (**g**). *n* = 6 biological replicates per group from two independent experiments. Box-and-whisker plots (min-to-max values with single data points). Two-tailed unpaired *t* test, *P*(hypothalamus) = 0.0024; *P*(rest brain) <0.0001. **i, j** ipGTT of mice fed a HFHSD for 4 weeks (wild-type C57Bl6J, **i**) or 8 weeks (wild-type Balb/c, **j**) that received vehicle or *i.v.* Treg transfers every 2 weeks. *N* = 4–6 biological replicates per group. Mean ± SD. Two-tailed unpaired *t* test, *P*(4 weeks) = 0.0531; *P*(8 weeks) <0.0001. Source data are provided as a Source Data file. *P < 0.05; **P < 0.01; ***P < 0.001; ****P < 0.0001.

weeks post-Treg transfer revealed that they had a significant reduction of HFHS-diet-induced hypothalamic immune activation such as local infiltration with macrophages (Fig. 8a, b). Analyses of microglia 2 weeks after Treg transfer in HFHS diet-fed mice indicated reduced immune activation as demonstrated by lower expression of costimulatory markers (CD80 and CD86, Fig. 8c, d). Moreover, hypothalamus-residing CD4+T cells in HFHS diet-exposed mice receiving Treg transfers likewise harbored significantly lower states of activation (Fig. 8f–h). In line with the reduced hypothalamic immune activation, Treg transfers supported an improved glucose tolerance (*P* = 0.0531) after 4 and 8 weeks (*P* < 0.0001) of HFHS diet (Fig. 8i, j).

In a fourth independent experimental approach, we performed bilateral intracerebral (*i.c.*)-transfers of Foxp3GFP+Tregs into the hypothalami of Balb/c mice fed the HFHS diet using bilateral injections and stereotactic surgery.

Immunofluorescence and confocal microscopy analyses of *i.c.*-transferred Foxp3GFP+Tregs confirmed the successful stereotactic delivery of Tregs directly to the hypothalamus (Fig. 9a). As a cellular control of transfer, control animals received bilateral injections of

equal numbers of naive non-activated CD4+CD25−CD44−T cells (for scheme see Supplementary Fig. 9d). After *i.c.* transfer of T cells, Balb/c mice were exposed to HFHS diet for 2 weeks.

Direct hypothalamic Treg transfers (for FACS sort purity of CD4+CD25+Foxp3GFP+Tregs for transfer, see Supplementary Fig. 9g) resulted in significantly reduced immune activation induced by HFHS diet (2 weeks) as indicated by analyses of frequencies (Fig. 9c) and phenotypes of infiltrating macrophages (Supplementary Fig. 9h–j). Accordingly, mice with hypothalamic Treg transfers showed reduced levels of immune activation as evidenced by lower expression of costimulatory markers (Fig. 9d–f). These findings were confirmed by analyses which demonstrated reduced numbers of MHCII/CD80 and MHCII/CD86 double-positive microglia upon Treg transfer in the presence of a HFHS diet (Supplementary Fig. 9k, l). Of note, direct *i.c.* transfer of Tregs into the hypothalamus supported a clear trend towards improved glucose tolerance (*P* = 0.0643, Fig. 9g) accompanied by trends toward reduced inflammatory NFκB-related genes in the adipose tissue (Supplementary Fig. 9e). These results are in agreement with the other Treg gain-of-function approaches as

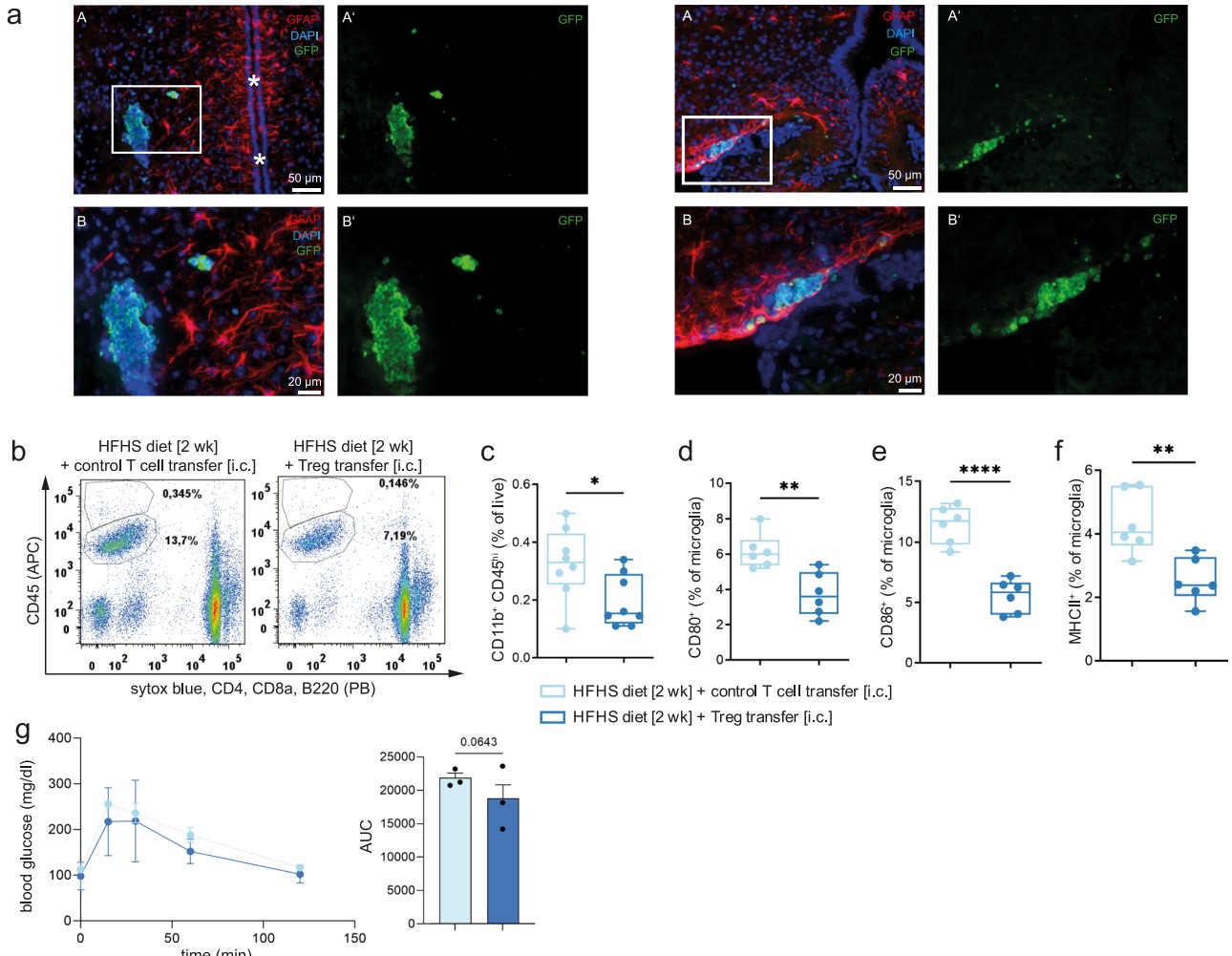

**Fig. 9 | Transferred Foxp3⁺Tregs reduce hypothalamic immune activation induced by the HFHS diet and improve metabolic indices. a** Localization of transferred CD4⁺CD25^high^Foxp3GFP⁺ Tregs after *i.c.* injection analyzed by fluorescence microscopy. Cryosections were stained for astrocytes (GFAP, red), nuclei (DAPI, blue) and GFP (transferred GFP+ Tregs, green). Localization of the *i.c.* transfer was shown in three independent experiments. Scale bar is 50 μm or 20 μm as indicated. **b** Representative FACS plots for CD45^int^CD11b⁺ microglia and CD45^hi^CD11b⁺ macrophages in brains from Balb/c mice fed the HFHS diet for 2 weeks. Prior to exposure to the HFHS diet, animals received *i.c.* Treg or cellular

control transfers. **c–f** Frequencies of CD45^hi^CD11b⁺ macrophages (**c**, $P = 0.0307$) and expression CD80⁺ ($P = 0.0037$), CD86⁺ ($P < 0.0001$), MHCII⁺ ($P = 0.0040$) cells in CD45^int^CD11b⁺ microglia (**d–f**). $n = 6$ biological replicates from two independent experiments. Box-and-whisker plots (min-to-max values with single data points). Two-tailed unpaired *t* test. **g** ipGTT of mice that received either *i.c.* control T cells or Tregs and were fed a HFHS diet for 2 weeks. Mean ± SD. $n = 3$ biological replicates per group. Two-tailed unpaired *t* test. Source data are provided as a Source Data file. $P < 0.05$; **$P < 0.01$; ***$P < 0.001$; ****$P < 0.0001$.

described above. Taken together, these series of gain-of-function experiments suggest that Tregs in the CNS have the potential to lower diet-induced immune activation and to improve systemic metabolism.

## Discussion

The hypothalamus is recognized as the key region regulating systemic metabolism[1,2]. Accumulating evidence indicates that exposure to calorie-rich diets rapidly promotes pathological neuronal changes, particularly in the hypothalamus, and that these changes might contribute to the development of obesity and diabetes. This neuronal dysfunction, however, is also associated with innate immune activation of microglia and infiltrating macrophages[5].

In line with our observations, recent studies suggest that the inhibition of microglial activation and expansion can distinctly reduce hypothalamic immune activation in response to a high-fat challenge[72]. Protein signatures of microglia from mice exposed to the HFHS diet revealed an involvement of adipocytokine signaling and highlighted innate activation characteristics, and likewise included a higher abundance of proteins including Ndufs2, Ndufs1 and Mapk1 that are

involved in Alzheimer's disease and which are particularly related to mitochondrial dysfunction. These findings are in agreement with a growing body of evidence suggesting that obesity and diabetes may increase the risk for developing cognitive impairment, including Alzheimer's disease and vascular dementia[73,74]. Future studies will be required to understand as to how and to what extent exposure to high-caloric diets accelerates microglial senescent degeneration[73,75,76].

Exposure of mice to the HFHS diet promoted a mild but significant upregulation of MHCII levels in residing microglia as well as upregulation of CD80 and CD86 in these cells, thereby highlighting their increased immune responsiveness resulting from the hypercaloric challenge. In addition, immunofluorescence analyses of Iba1⁺ microglia of mice fed the HFHS diet revealed high levels of MHCII near the choroid plexus and median eminence as well as the meninges. In addition, Iba1 expression of macrophages might contribute to the observed MHCII expression in antigen-presenting cells since these immunofluorescence analyses do not distinguish between CD45^high^ vs. CD45^intermediate^ cells. We confirmed the findings on MHCII expression by analyses with CD45^int^CD11b⁺ microglia from Cx3cr1^GFP^ reporter mice,

suggesting that an upregulation of MHCII expression in microglia does indeed contribute to the observed results upon hypercaloric challenge. Only recently, a fluorescent microglia reporter mouse was described that uses a binary split Cre system, where the co-expression of Iba1 and Cx3cr1 is required to induce the transcription of a fluorescent reporter[77]. This novel reporter mouse might help to decipher the contribution of the aforementioned cell populations in response to hypercaloric challenges in further depth.

An adaptive immune response, and particularly the recruitment of T cells to the CNS, has long been considered pathogenic, especially in the context of neuroinflammatory diseases. However, recent findings have also described several functions for T cells in the healthy CNS, including a critical impact on microglia maturation, proper synaptic pruning and behavior[32,34,36]. While studies in healthy humans identified only sporadic presence of T cells in the brain parenchyma[78], around 150,000 T lymphocytes were identified in human CSF[79]. These cells were found to patrol the CSF or perivascular space for detrimental antigens in the healthy mouse[80], thereby supporting surveillance of the local environment.

A detailed analysis of the anatomical localization of specific subsets of antigen-presenting cells will improve our understanding of how T-cell responses develop in distinct parts of the CNS following environmental challenges such as exposure to HFHS diets. In this regard, recent studies have highlighted the dural sinuses as a unique interface of CNS and immune system communication[29]. At this location, CNS-antigen exposure, drainage via the CSF, uptake by APCs and presentation to T cells are functional to support immune surveillance. Therefore, future studies will be required to dissect whether exposure to calorie-rich diets will modulate the immunogenicity of local antigens and/or trigger the presentation of neo-epitopes, thereby supporting local immune activation.

Here, we define CD4+T cells, including specifically Foxp3+Tregs, as critical players in a cellular interplay involving astrocytes, microglia and infiltrating macrophages, to maintain hypothalamic immuno-metabolic integrity in response to hypercaloric challenges. CNS immune crosstalk involves Foxp3+Tregs to functionally interact with neuronal networks. RNA-seq data of hypothalamic CD4+T cells following a high-caloric challenge indicate a Th1-like activation state as revealed by high expression of *Tbx21*, *Ifng*, *Cxcr3* and *Cd226*[52]. Moreover, locally-residing CD4+T cells present with upregulated *Ccl5* as well as reduced expression of *Ccr7* and *S1pr1*, supporting their recruitment to inflammatory tissues as well as their retention at these sites[29,48]. Accordingly, after a chronic high-caloric challenge, we observed a mild increase in local CD4+T-cell frequency. In support of these observations, previous analyses in the setting of autoimmunity such as multiple sclerosis support the notion that very low numbers of specific T cells can control inflammatory responses in the target organ[81].

Regarding T-cell entry into the brain, cellular transmigration across the blood-brain barrier is not a single-step process as observed in other tissues, and more importantly involves two distinct processes that overlap within the perivascular space[82,83]. Recent studies suggest that under physiological conditions, the dissemination of T cells within the CNS is limited to the CSF-filled ventricular and subarachnoid spaces, as well as dural sinuses[29,84,85]. Following intraparenchymal damage or inflammation, the glia limitans becomes permissive by the action of matrix metalloproteinases (MMPs) MMP2 and MMP9, thereby permitting migration of T cells into the neuropil. During neuroinflammation the exchange of blood-derived cells within the perivascular space would provide opportunities for interaction of T cells with MHC-matched antigen-presenting cells[29,59,85]. Accordingly, T-cell localization will impact possibilities for interacting with antigen-presenting cells and thereby can contribute to the promotion of immune activation and local inflammatory settings. It will be of interest to understand whether additional mechanisms will permit T-cell entry under normal physiologic conditions to support tissue-specific immune surveillance and which stimuli and/or local inflammatory signals promoted by exposure to HFHS diets will modulate permissiveness of the glia limitans for immune cell entry[86].

We are only beginning to understand the role of tissue-resident T cells in immune surveillance of the CNS[29,32,36–38,42,81] and additional studies are required to advance our understanding of how these cells are modulated or triggered upon exposure to calorie-rich diets. Here, we show low-frequency CD4+T cells including Foxp3+Tregs in the parenchyma of brains of young lean healthy mice. These observations are in line with recent findings[7,36,38,43] and support the notion that in addition to their localization within the perivascular compartment, a small number of CD4+T-cell populations and Foxp3+Tregs also reside in the neuropil in the steady state. Moreover, recent studies have highlighted that T cells residing in the perivascular space secrete soluble mediators to critically impact and crosstalk with neighboring cells including neurons, astrocytes and microglia[31,32,34]. Additional studies will be required to dissect the role of these T cells in supporting the normal function of the immune system and their potential aberrant activation upon metabolic challenges such as the HFHS diet. Another important aspect that will require further investigation in the future is the timing of the observed effects, particularly concerning the onset and progression of phenotypic changes in immune cell populations (i.e., Tregs, microglia and macrophages). While the activation of macrophages and microglia in response to hypercaloric feeding can be seen as early as 1 week after exposure to the HFHS diet, the reduction of Tregs in the hypothalamus occurs more gradually. Specifically, we identify a modest but significant reduction in Tregs in the hypothalamus after 8 weeks of HFHS diet. Despite the observed decline in hypothalamic Tregs in response to HFHS diet, the assessment of any functional impairments in local Tregs that are caused by this hypercaloric challenge remains currently experimentally unfeasible, since only very few cells can be isolated from such hypothalami of mice. Due to these experimental limitations, it is currently not possible to draw precise direct conclusions on whether Tregs in the hypothalamus exhibit a different functional profile/polarization/signature upon exposure to the HFHS diet and prior to their decline in the hypothalamus.

To address the question whether the hypercaloric environment affects the whole Treg population in general, we used analyses from peripheral lymph node derived Tregs. These analyses from peripheral lymph nodes showed that a HFHS diet per se does not have a major impact on Treg functionality. A direct assessment of hypothalamic Treg functionality is experimentally not possible currently, as highlighted above. Therefore, we used a series of independent Treg depletion strategies followed by exposure of animals to a hypercaloric challenge to dissect the functional relevance of Tregs in this anatomical niche. These Treg depletion experiments revealed a significant increase in frequencies and phenotypes of infiltrating macrophages. Moreover, microglial immune responsiveness was distinctly enhanced in Treg-depleted mice when compared to a hypercaloric-challenge in Treg-complete mice. Vice versa, peripheral Treg transfer significantly abrogated infiltration with macrophages and their activation and furthermore, significantly reduced microglial reactivity in mice challenged with the HFHS diet. A direct involvement of Foxp3+Tregs in limiting immune activation and responsiveness given high-caloric feeding is revealed based on *i.c.* Treg transfers which significantly lowered local immune activation and reactivity of macrophages and microglia.

Conceptually, the observed decline in hypothalamus-residing Foxp3+Tregs following a high-caloric challenge, together with an increase in local T-cell activation indicates that a specific tissue microenvironment instils functional specialization of Tregs in that particular niche to optimize local immune regulation, function and tissue homeostasis. In line with this view, adipose tissue Tregs control local tissue integrity, function and inflammation[18–20,44,87]. Accordingly,

it is interesting to speculate that such functional adaptation of tissue Tregs in distinct metabolic tissues including brain, adipose tissue, liver or muscle[88,89] supports the integration of immune-metabolic crosstalk in these distinct compartments[90]. As one example, leptin and other adipokines released by adipocytes signal through the hypothalamus to control systemic metabolism, including modulation of sympathetic output to adipose tissue[91,92]. Therefore, it will be of relevance to dissect in future studies whether local Tregs in the brain can impinge not only on the polarization and signaling of immune cells as adipose-residing macrophages but also on other cells as adipocytes and sympathetic nerves[92,93]. Accordingly and from a Treg targeting perspective, we found that specific Treg expansion reduced NFκB-related target genes in the hypothalamus, lowered inflammatory characteristics in the adipose tissue and improved systemic metabolic parameters.

Moreover, work from Bapat et al.[94] suggested that mice with age-associated insulin resistance benefit from a reduction in adipose tissue Tregs upon depletion of these cells using anti-ST2 antibodies. Therefore, it will be of interest in future studies to assess the implications of Tregs in the CNS in modulating age- vs. obesity-associated immune-metabolic alterations in the adipose tissue.

Overall, the findings presented herein might open new avenues for the future design of tailored concepts to manipulate Treg-based immuno-metabolic crosstalk in diet-induced obesity and T2D[90].

## Limitations of the study

One important limitation of the study is that specific Treg targeting approaches available in the field[71,94] as expansion by anti-IL2/IL2 antibody complexes will also modulate peripheral Treg populations e.g., in LNs or in other tissues. Therefore, to dissect a direct functional role of Tregs in the hypothalamus, we performed also i.c. Treg transfers. However, results here might be possibly confounded by local micro injury induced by i.c. transfers, albeit we used non-Tregs as a cellular i.c. transfer in control animals. Application of anti-IL2/IL2 antibody complexes into the 3rd ventricle using osmotic mini-pumps or targeting other brain-resident cells such as astrocytes and inducing local IL2 expression[43] might be an interesting possibility of future local Treg expansion in the context of diet-induced obesity and T2D.

## Methods

### Mice

Balb/cByJ, C.Cg-*Foxp3*[tm2Tch]/J, B6.129P-*Cx3cr1*[tm1Litt]/J, C57Bl/6J, B6.Cg-*Lep*[ob]/J, B6;129-*Gt(ROSA)26Sortm2Sho*/J, B6.Cg-Tg(CD4-Cre)1Cwi/BfluJ, C57BL/6-Tg(Foxp3 DTR/EGFP)23.2Spar/Mmjax, B6.SJL-*Ptprc*[a]*Pepc*[b]/BoyJ and B6.Cg-*Foxp3*[tm2Tch]/J were obtained from The Jackson Laboratory and maintained by in-house breeding at the animal facility of the Helmholtz Zentrum München, Munich, Germany, according to guidelines established by the Institutional Animal Committees. CD4GFP mice were obtained from crossbreeding of CD4-Cre mice with Rosa loxP stop loxP eGFP mice. Cre-mediated excision of the floxed stop codon led to the expression of GFP in cells that expressed CD4. C57BL/6-Tg(Foxp3 DTR/EGFP)23.2Spar/Mmjax, referred to as Foxp3 DTR mice, were kindly provided by Prof. Dr. Tobias Bopp, Institute of Immunology, University Medical Center Mainz, Johannes Gutenberg-University, Mainz, Germany, and maintained at the animal facility of the I. Medical Clinic, University of Erlangen-Nuremberg, Erlangen, Germany or at the Helmholtz Zentrum München, Munich, Germany.

All mice were group-housed on a 12 h/12 h light−dark cycle at 22–24 °C under specific pathogen-free (SPF) conditions in individually ventilated cages. Mice had ad libitum access to food (irradiated standard diet (Altromin, #1314, Lage, Germany) or high-fat high-sugar (HFHS) diet where indicated (Research Diets, #D12331i, New Brunswick, NJ)) and sterilized water. Hypercaloric challenges using HFHS diet were started using 6-week-old female Balb/c mice or male C57Bl/6J mice. Where indicated, mice were perfused transcardially with 0.9% NaCl and 10 U/ml heparin for 5–8 min after ketamine/xylazine overdose.

Ethical approval for all mouse experimentations has been received by corresponding local animal welfare authorities (District government of Upper Bavaria, Germany, #ROB-55.2-2532.Vet_02-17-63, #ROB-55.2-2532.Vet_02-18-173, #ROB-55.2-2532.Vet_02-21-196 and #ROB-55.2-2532.Vet_02-23-116).

Mice were randomized to test groups. Whenever possible, littermates were used. Animals were not involved in previous procedures. No animals were excluded due to illness or outlier results; therefore, no exclusion determination was required. The investigators were not blinded to group allocation or to the assessment of experimental end points.

See also Supplementary Table 2 for further information about all mouse strains used.

### In vivo cold exposure

For cold exposure experiments, cages were placed at 8 °C ambient temperature for 1 week and mice remained group-housed with a 12 h/12 h light−dark cycle.

### In vivo Treg depletion

For loss-of-function experiments, Tregs were depleted by i.p. injection of 250 μg anti-mCD25 antibodies (BioXcell, clone PC-61.5.3) or vehicle on 3 consecutive days. In a second approach, male Foxp3 DTR (DEREG) mice were treated with 50 ng diphtheria toxin (DTX) per gram body weight on 3 consecutive days. Male mice negative for Foxp3 DTR (DTR⁻) were used as control and also received DTX.

### Intracerebral T-cell transfer

For the gain-of-function experiment, either 100,000 CD4⁺CD25ʰⁱFoxp3GFP⁺ Tregs or naive CD4⁺CD25⁻CD44ˡᵒʷFoxp3GFP⁻ T cells were FACS-sorted and injected i.c. into the arcuate nucleus of recipient mice using a stereotactic device. Mice were anesthetized via i.p. ketamine (100 mg/ml) and xylazine (10 mg/ml), followed by s.c. 200 mg/kg metamizol. The skin on the head was shaved and sterilized with 70% ethanol and 10% povidone iodine. An incision was made ~1 cm with a sterile scalpel. The coordinates were set using a stereotactic frame. Two holes were drilled for bilateral injections and T cells were injected in a total volume of 1 μl over 5 min (coordinates relative to bregma: −1.3 mm anterior-posterior and ±0.3 mm medial-lateral). The Hamilton needle was removed over 5–10 min. Mice were sutured and allowed to recover on a heating pad. Post-surgical analgesia was daily s.c. meloxicam (1 mg/kg) for 3 days. Here, hypothalami and brains were analyzed after 1–2 weeks on the HFHS diet.

### Treg expansion using anti-IL2/IL2 antibody complexes

For Treg expansion experiments, mice were injected initially on 3 consecutive days with 6 μg anti-IL2/IL2 Ab complexes (or vehicle) as also described earlier[71]. On day 5, the peak of the expansion, the HFHS diet was initiated, and mice were injected twice weekly until the end of the observation period. The tail vein was punctured weekly for blood glucose measurements and flow cytometric tracking of T-cell/Treg populations in the blood.

### IV Treg transfer

For i.v. Treg transfers, live CD4⁺CD25ʰⁱFoxp3 GFP⁺ Tregs were FACS-sorted for purity on the BD FACS Aria III. 500,000 Tregs were injected i.v. in sterile PBS. For multiple i.v. Treg transfers, Foxp3 GFP⁺ Tregs were enriched using the Miltenyi regulatory T-cell enrichment kit based on magnetic sorting of CD4⁺ CD25ʰⁱ Tregs. Post-sort purity controls are shown in the supplement.

### Intraperitoneal glucose tolerance test (ipGTT)

ipGTTs were performed according to standard procedures. In short, mice were fasted for 6 h (8 am until 2 pm) with free access to water. After basal glucose measurements, mice were injected with 2 g/kg body weight D-glucose in saline (Carl Roth). Blood glucose was

measured at the indicated time points, and the area under the curve (AUC) was calculated using GraphPad Prism.

## Isolation of cells from mouse brain

For mouse brain isolation, the skull was carefully opened using scissors, carefully cutting clockwise. Isolated brains were stored in Hanks Balanced Salt Solution (HBSS, Sigma Aldrich) supplemented with 5% FCS and 10 mM HEPES (referred to as HBSS⁺) on ice. Olfactory bulbs and the cerebellum were removed using a scalpel. The hypothalamic area of the brain was carefully excised using fine Dumont forceps.

Cells from brains and hypothalami were isolated by homogenization in RPMI media with a hand homogenizer on ice, followed by density gradient centrifugation with 30% vs. 70% Percoll for 30 min at 500×$g$ (acceleration 1, deceleration 1, room temperature). The interphase containing microglia and lymphocytes was collected, and cells were washed with HBSS⁺ prior to further processing.

## Magnetic enrichment of brain cells

Pelleted cells, collected from interphases of the density gradient centrifugation, were magnetic-activated cell-sorting (MACS)-enriched using CD11b microbeads (Miltenyi) and LS (brain) or MS (hypothalamus) columns according to the manufacturer's instructions, followed by staining for flow cytometric analysis. Microglia were gated as dump⁻CD45ⁱⁿᵗCD11b⁺, while co-purified macrophages were gated as dump⁻CD45ʰⁱCD11b⁺. The flow through contained all CD11b⁻ cells lymphocytes which were used for further flow cytometric analyses of T cells.

For MACS sorting of astrocytes from brains and hypothalami, the ACSA2+ Kit from Miltenyi was used according to the manufacturer's instructions with MS columns.

## Isolation of T cells from lymphoid organs

Lymph nodes were mashed through 70-μm cell strainers in HBSS⁺. Single-cell suspensions were kept on ice and further processed for flow cytometric analysis and sorting as described below.

## T-cell stimulation for cytokine analyses

T cells were stimulated in complete RPMI with 0.5 μg/ml PMA and 0.5 μg/ml ionomycin in the presence of 1.5 mM calcium chloride in a humidified CO₂ incubator. The cells were stimulated for 4 h and after the initial 2 h, the brefeldin A containing inhibitor GolgiPlug (Beckton Dickinson) was added 1:1000.

## Flow cytometry and cell sorting

Unspecific binding of antibodies was prevented by incubation of single-cell suspensions with Fc-Block (BD Pharmingen, 2.4G2, 1:100) in HBSS⁺ for 10 min on ice, followed by surface staining for 30 min on ice in the dark. All monoclonal antibodies used for mouse FACS stainings are listed in Supplementary Table 1. Before acquisition on the flow cytometer, cells were passed through a 40 μm strainer (NeoLab) to remove large debris.

To detect intracellular protein expression (e.g., of Foxp3 in cells of non-reporter mice or Ki67), T cells were fixated and permeabilized using the Foxp3 Staining Buffer Set (eBioscience) after surface staining. Intracellular proteins were stained for 30 min on ice in the dark and washed 2–3 times with Perm buffer and 1–2 times with HBSS+ before acquisition on the flow cytometer.

For cytokine expression analyses, intracellular antigens were stained overnight at 4 °C after stimulation and fixation as described above.

Cells were acquired on a BD FACSAriaIII flow cytometer using FACSDiva software with optimal compensation and gain settings determined for each experiment based on unstained and single-color stained samples. Fluorescence minus one (FMO) controls were used to set gates. Doublets were excluded based on SSC-A vs. SSC-W plots and

FSC-A vs. FSC-W plots. Live cell populations were gated on the basis of cell side and forward scatter and the exclusion of cells positive for Sytox Blue, Sytox Red (both Life Technologies) or Fixable Viability Dye eFluor450 (eBioscience). Samples were analyzed using FlowJo software version 7.6.1 or 10.7.1 (TreeStar Inc., OR).

## mRNA expression profiling

Pooled sample-sets of FACS-sorted CD4⁺T cells (see figure legends for full gating strategy) purified from mice on standard diet vs. HFHSD and *ob/ob* vs. C57Bl/6J mice were used for Next Generation Sequencing (NGS). DNA was generated using the SMARTer ultra-low input RNA Kit for sequencing−v4 or SMARTer® Universal Low Input RNA Kit for Sequencing (Takara Clontech) according to the manufacturer's instructions. cDNA concentrations for NGS were assessed using Agilent High Sensitivity DNA Chips (Agilent Technologies) and Agilent 2100 Bioanalyzer. mRNA library preparation was conducted with Nextera reagents (Illumina) according to the manufacturer's instructions. NGS was performed on a NextSeq (Illumina) with 75 bp single-end reads for mRNA using Illumina reagents and following the manufacturer's instructions. Quality was assessed by FastQC[95]. Reads were mapped to the mouse genome (mm10) using BWA-mem with default configuration. Read count lists were created using SAMTools[96] and HTSeq-count[97]. Reads were normalized using DESeq2[98]. Normalized read counts were further processed for descriptive visualization of expression trends and overlaps using R/bioconductor[99,100]. Gene lists were enriched with Gene Ontology terms[101,102] using DAVID Bioinformatics Resources 6.7[103]. For further analysis and graphical presentation, the DESeq2-normalized counts were used to calculate the fold change of HFHS diet/SD T cells. A cutoff of >20 reads and a fold change of >2 or <0.5 resulted in 1240 regulated genes for Treg analyses and 2791 genes for T cells.

During the revision of this manuscript, the RNA sequencing of conventional T cells (gated as live CD4⁺ CD25⁻ Foxp3GFP⁻ CD44ʰⁱ CD62Lˡᵒʷ) was repeated. Pooled cells (from five) mice on standard diet vs. HFHSD were used for each sample. The SMARTer mRNA+LP Kit for Sequencing (Takara Clontech) was used according to the manufacturer's instructions. Sample concentrations were determined by Qubit and Agilent 2100 Bioanalyzer. NGS was performed on a NovaSeq X Plus (Illumina) with 100 bp paired-end reads using Illumina reagents on three lanes. Raw FASTQ files were processed using the nf-core/rnaseq pipeline (version 3.16) (DOI: 10.5281/zenodo.1400710) with default parameters, using the mm10 mouse reference genome. Gene counts were quantified with Salmon[104] and subsequently analyzed for differential gene expression with DESeq2[99]. A cutoff of >10 reads was applied. Heatmaps were generated using the pheatmap package[105], based on variance-stabilized transformed counts from the 500 genes exhibiting the highest variance across samples. For gene set enrichment analysis, the fGSEA[106] package was employed, utilizing gene sets from MSigDB (Molecular Signatures Database)[107] and KEGG[108–110].

## Quantitative real-time PCR

For validation of NGS experiments, CD4⁺CD25⁻CD44ˡᵒʷ⁺ⁱⁿᵗ T cells were FACS-sorted for two-way purity. RNA and cDNA were isolated and pre-amplified using the SMARTer ultra-low input RNA Kit for sequencing v4 (Takara Clontech). Quantitative real-time PCR was performed using the Ssofast Real Time Mix (BioRad) and gene expression was normalized to *Histone* as housekeeping gene.

For RNA isolation from total tissue, the Beckman Coulter RNAdvance Tissue Kit was used according to the manufacturer's instructions. RNA was isolated from ½ rest brain, ½-1 hypothalamus, ~5 mg adipose tissue including the optional DNase I digestion. 1 μg of RNA was reverse transcribed using the iScript Advanced kit. Real-time PCR was performed using the Ssofast Real Time Mix (BioRad).

For sorted cell populations, the Beckman Coulter RNAdvance Cell v2 Kit was used. RNA was reverse transcribed using the iScript

Advanced Kit (BioRad) and real-time PCR was performed using the Ssofast Real-time Mix (BioRad). Gene expression was normalized to *Histone H3*. For primer sequences, see Supplementary Table 3.

## Proteomics

Mice were fed either standard diet or a HFHS for 36 weeks. Microglia were isolated as described above and sorted for purity as live CD45[int]CD11b[+] on a FACSAriaIII cell sorter (BD). **Sample preparation for LC-MS/MS analysis**: Cell lysis of FACS-sorted cells was performed in LB buffer (50% (v/v) 2-2-2-trifluoroethanol (TFE) plus 2 mM dithiothreitol, 50% (v/v) 50 mM ammonium bicarbonate (ABC) buffer) at 99 °C for 10 min followed by sonication for 10 min (10 cycles high intensity, Bioruptor, Diagenode). Cell debris was removed after 10 min centrifugation (16,000× *g* at 4 °C), and proteins in the lysate were alkylated for 30 min with 10 mM iodoacetamide in the dark. Next, the solution was centrifuged in a vacuum evaporator for about 3 h at 45 °C. Proteins were resolubilized in 10% (v/) TFE in 50 mM ABC and sonicated in a water bath for 5 min. Proteins were digested by adding 0.2 μg of LysC and 0.2 μg of Trypsin and incubation at 37 °C overnight. The next day, the digest was stopped by adding 1% (v/v) TFE and the solution volume was reduced in a vacuum evaporator for about one h at 45 °C. Samples were finally desalted on SDB-RPS StageTips (3 M, Empore, Neuss, Germany) and eluted as described (Kulak et al., 2014). **LC-MS/MS analysis**: MS analysis was performed using Q-Exactive HF mass spectrometers (Thermo Fisher Scientific, Bremen, Germany) coupled online to a nanoflow UHPLC instrument (Easy nLC, Thermo Fisher Scientific). Peptides were separated on a 50-cm long (75 μm inner diameter) column packed in-house with ReproSil-Pur C18-AQ 1.9-μm resin (Dr. Maisch GmbH, Ammerbuch, Germany). Column temperature was kept at 50 °C by an in-house designed oven with a Peltier element and operational parameters were monitored in real time by the SprayQc software[111]. Peptides were loaded with buffer A (0.1% (v/v) formic acid) and eluted with a nonlinear gradient of 5–60% buffer B (0.1% (v/v) formic acid, 80% (v/v) acetonitrile) at a flow rate of 250 nl/min. Peptide separation was achieved by 100-min gradients. The survey scans (200–2000 *m/z*, target value = 3E6, maximum ion injection times = 20 ms) were acquired at a resolution of 60,000 followed by higher-energy collisional dissociation (HCD) based fragmentation (normalized collision energy = 27) of up to 15 dynamically chosen, most abundant precursor ions (isolation window = 1.4 *m/z*). The MS/MS scans were acquired at a resolution of 15,000 (target value = 1E5, maximum ion injection times = 60 ms). Repeated sequencing of peptides was minimized by excluding the selected peptide candidates for 20 s. Computational MS data analysis—All data were analyzed using the MaxQuant software package 1.5.3.29[112]. The false-discovery rate (FDR) cutoff was set to 1% for protein and peptide spectrum matches. Peptides were required to have a minimum length of 7 amino acids and a maximum mass of 4600 Da. MaxQuant was used to score fragmentation scans for identification based on a search with an initial allowed mass deviation of the precursor ion of a maximum of 4.5 ppm after time-dependent mass calibration. The allowed fragment mass deviation was 20 ppm. Fragmentation spectra were identified using the UniprotKB *Mus musculus* database[113], based on the 2014_07 release, combined with 262 common contaminants by the integrated Andromeda search engine[114]. Enzyme specificity was set as C-terminal to arginine and lysine, also allowing cleavage before proline, and a maximum of two missed cleavages. Carbamidomethylation of cysteine was set as fixed modification and N-terminal protein acetylation as well as methionine oxidation as variable modifications. Both 'label-free quantification (MaxLFQ)' and 'match between runs' with standard settings were enabled[115]. Protein copy number estimates were calculated using the iBAQ algorithm[116]. **Statistics and Data visualization**: Basic data handling, normalization, statistics and annotation enrichment analysis were performed with the Perseus software package[117]. We filtered for 3222 protein groups that were quantified with at least

two valid values in at least one group of triplicates. Missing values were imputed with values representing a normal distribution (generated at 1.8 standard deviations of the total intensity distribution, subtracted from the mean, and a width of 0.3 standard deviations). Differentially expressed proteins were identified by one-way ANOVA test at a permutation-based FDR cutoff of 0.05. Enrichment for annotation categories was evaluated by 1D annotation enrichment or Fisher exact test to obtain a *P* value. The mass spectrometry proteomics data have been deposited to the ProteomeXchange Consortium[118] via the PRIDE partner repository with the dataset identifier PXD004498.

## Fluorescence microscopy

For cryostat sections, brains were removed and postfixed in 4% paraformaldehyde (Sigma Aldrich, Steinheim, Germany) for 24 h. For cyro protection, the tissue was incubated in 10%, 20% and 30% sucrose (Merck, Darmstadt, Germany) in PBS, each for 24 h. Next, the tissue was transferred in cryo embedding medium (Sakura Finetek, Torrance, USA), frozen and consecutively cut with a cryo microtome (Leica Microsystems, Wetzlar, Germany) at 14 μm thickness. Sections were blocked and permeabilized in PBS containing 3% normal donkey serum (DKS) (Sigma Aldrich) and 0.3% Triton X100 (Roth, Karlsruhe, Germany) for 15 min at room temperature. Primary antibodies (goat anti-GFP, 1:200, Acris antibodies, Herford, Germany and rabbit anti-GFAP, 1:500, Dako, Glostrup, Denmark) were incubated overnight at 4 °C followed by thorough rinsing and incubation with appropriate secondary antibodies (donkey anti-goat Alexa Fluor 488 and donkey anti-rabbit Alexa Fluor 568, both 1:250, Life Technologies, Eugene, Oregon, USA) for 2 h at room temperature) in PBS containing 1% DKS and 0.03% Triton X100.

For analysis using vibratome sections, animals were sacrificed and transcardially perfused with saline followed by 4% paraformaldehyde (Sigma Aldrich) in PBS. After removal of the brains, the tissue was postfixed in the same fixative overnight. Next, vibratome sections were acquired at a thickness of 30 μm. Prior to immunolabeling, the sections were permeabilized by a short cycle of freezing and thawing in PBS. Sections were then blocked and further permeabilized with 3% DKS and 0.5% Triton X100 in PBS for 30 min. Primary antibodies directed against GFP (goat anti-GFP, 1:200, Acris antibodies) and Iba1 (rabbit anti-Iba1, 1:200, Synaptic Systems, Göttingen, Germany) were incubated for 2 h at room temperature under slight agitation. After thorough rinsing in PBS, appropriate secondary antibodies (donkey anti-goat Alexa 488 and donkey anti-rabbit Alexa 568, both 1:250, Life Technologies) were incubated for 2 h.

In cryostat as well as vibratome sections, nuclei were counterstained with 4′,6-diamidino-2-phenylindole dihydrochloride (DAPI) (Sigma Aldrich, 1:10,000) for 5 min.

Consecutive sections were analyzed with an Olympus BX51 epifluorescence microscope equipped with an XM10-cooled CCD grayscale camera and Cell Sens software (Olympus, Hamburg, Germany).

As a second approach for cryosections, brains were isolated, fixed and cut by microtome in 30 μm sections as described before. After permeabilization with TritonX100 for 5 min and incubation in protein block (DAKO) with 2% BSA for 10 min sections were incubated with primary antibodies (Armenian Hamster−anti-mCD3, clone 145-2C11 (Biolegend) and rat-anti-mFoxp3, clone FJK-16s, eBioscience) overnight at 4 °C followed by washing step and incubation with an appropriate secondary antibodies (goat anti-hamster conjugated with AlexaFluor[488] (Jackson Lab) and biotinylated goat anti-rat followed by use of TSA-Cy3 Amplification Kit (Perkin-Elmer) for 2 h at RT. For analysis of the antigen CD4 and Collagen IV, cryosections were incubated with the primary antibodies (rat-anti-mCD4, clone GK1.5 (BD) and rabbit anti-mCollagenIVType1, AB765P (Millipore)) overnight at 4 °C and then incubated with secondary antibodies (biotinylated goat anti-rat (Dianova) followed by use of TSA-Cy3 Amplification kit (Perkin-Elmer) or goat anti-rabbit conjugated with Alexa Fluor 488 dye (Jackson) for 2 h at RT. For analysis of the antigen CD4 and GFAP,

cryosections were incubated with primary antibodies CD4 as described above and polyclonal rabbit anti-mGFAP (DAKO) overnight at 4 °C followed by goat anti-rabbit conjugated with AlexaFluor 488 (Jackson) for 2 h at RT. Nuclei were counterstained with DAPI (Vector) and sections were finally analyzed by fluorescence microscopy (Olympus).

### Light microscopy

For light microscopy, vibratome sections were acquired as described above. Prior to immunolabeling, endogenous peroxidases were blocked by incubation in methanol containing 0.3% hydrogen peroxide for 30 min at room temperature. Next, the tissue was blocked and permeabilized with normal rabbit serum and 0.3% Triton X100 in Tris-buffered saline (TBS). The primary antibody (goat anti-GFP, 1:200, Acris antibodies) was incubated for 2 h under slight agitation, followed by thorough rinsing in TBS. The biotinylated secondary antibody (1:250, rabbit anti-goat, 1:250, Vector Laboratories, Burlingame, CA, USA) was incubated for 2 h at room temperature under slight agitation. After thorough rinsing, the streptavidin peroxidase complex was allowed to bind to the biotinylated antibodies (ExtrAvidin, 1:100, Sigma Aldrich) for 1 h followed by thorough rinsing and diaminobenzidine (DAB) (Sigma Aldrich)-mediated staining. After final rinsing, the sections were transferred onto microscope slides, air-dried and embedded under cover glasses in Entellan (Merck).

Images were acquired using a Zeiss Axioskop (Carl Zeiss Microscopy, Jena, Germany) equipped with a Progres-cooled CCD camera (Jenoptik, Jena, Germany).

### Statistics and reproducibility

Results are presented as mean and standard error of the mean (mean ± SEM), mean and standard deviation (mean ± SD) or as percentages, where appropriate. Summary graphs of data are presented using box-and-whisker plots indicating minimum to maximum values with all data points to demonstrate data distribution or bar graphs (with single data points overlaid), as well as single data points. For normally distributed data, two-tailed Student's $t$ test for unpaired values was used to compare means between independent groups. For more than two groups, an ordinary one-way ANOVA was used with Šidák post hoc test for multiple comparison. For all tests, a two-tailed $P$ value of <0.05 was considered to be significant. Statistical significance is shown as $*P < 0.05$; $**P < 0.01$; $***P < 0.001$; $****P < 0.0001$. Analyses were performed using the programs GraphPad Prism 6.0.1 and 9.5.1 (La Jolla, CA) and the Statistical Package for the Social Sciences (SPSS 19.0; SPSS Inc., Chicago, IL).

No statistical method was used to predetermine sample size. Mice were randomized to test groups. The Investigators were not blinded to allocation during experiments and outcome assessment. No animals were excluded due to illness or outlier results; therefore, no exclusion determination was required. The presence of CD4+ T cells in the healthy mouse brain was confirmed by experiments/stainings/analyses in three independent labs. For FACS analyses, experimentators were not blinded to experimental groups during in vivo experiments (handling, scoring etc.), but during sample processing and data acquisition. For microscopic analyses, experimenters were blinded to groups. For proteome analyses, experimentators were blinded during sample processing and acquisition.

### Reporting summary

Further information on research design is available in the Nature Portfolio Reporting Summary linked to this article.

### Data availability

All data supporting the results of this study are available in the article, Supplementary Information and/or Source Data file, provided with this paper. The mass spectrometry proteomics data generated in this study have been deposited to the ProteomeXchange Consortium (Vizcaino et al.[118]) via the PRIDE partner repository with the dataset identifier PXD004498. The sequencing data generated in this publication have been deposited in NCBI's Gene Expression Omnibus[119] and are accessible through GEO Series accession number GSE282067. Source data are provided with this paper.

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

## Acknowledgements

The authors thank Vigo Heissmeyer, Ludwig-Maximilians-Universität München, Munich, Germany for providing primer sequences for NF-kB-related analyses and Maria Rohm and Julia Geppert, both Institute for Diabetes and Cancer, Helmholtz Zentrum München, Munich, Germany for plasma analyses. C.D. holds a professorship grant from the Excellence Program for Outstanding Female Scientists from the Helmholtz Association, is supported by a Research Division at Helmholtz Zentrum München, the German Center for Diabetes Research (DZD), through the Deutsche Forschungsgemeinschaft (DFG, German Research Foundation)– Project-ID 210592381—SFB 1054 (B11), project number 490846870—TRR355/1 TPA02 and through an award of the EFSD/JDRF/Lilly Programme on Type 1 Diabetes Research 2020. MB is supported by the DZD. I.S. is supported by the DFG, project number 490846870—TRR355/1 TPB02 and by the DZD. A.W. is supported by project number 490846870—TRR355/1 TPA08 and TPB05. T.B. is supported by project number 490846870—TRR355/1 TPA09 and TPA10. REK was supported by the Wilhelm-Sander-Stiftung (2013.111.1), the Anni-Hofmann-Stiftung and the „Verein zur Förderung von Wissenschaft und Forschung an der Medizinischen Fakultät der LMU München" (WiFoMed). B.W. is supported by WE 4656/2 and DFG-CRC1811 (B02). M.H.T. was supported by the Alexander von Humboldt Foundation, ERC HypoFlam (#695054), the DZD, and the Helmholtz Alliance ICEMED-Imaging and Curing Environmental Metabolic Diseases, through the Initiative and Networking Fund of the Helmholtz Association. Proteomics work at the Max Planck Institute of Biochemistry was supported by the Max Planck Society for Advancement of Science. The authors acknowledge the technical support of the Core Facility Laboratory Animal Services (CF-LAS) and Core Facility Genomics (CF-GEN) at Helmholtz Munich.

## Author contributions

M.B. performed experiments, analyzed data, prepared figures and wrote the manuscript; S.K. and A.H.N. performed experiments, analyzed data; V.K.S. helped with ex vivo analyses; V.B.O. prepared T cells for the first set of NGS analyses; R.E.K. and B.L. performed gain-of-function studies; M.G.S. supported data presentation and prepared supplemental figure 10; L.R.N. supported transfer experiments; I.S., D.O., and H.H. supported in vivo experiments; M.H. and M.L. performed analyses of NGS data; F.H. performed proteomic profiling of microglia; M.W.P. supported NGS data analyses; M.K. and B.F. performed IF and IHC experiments; A.N. and A.M. performed IF and IHC studies; A.W. supported IF and IHC studies; B.H. performed sequencing analyses; C.X.Y. supported microglia analyses; S.D. characterized Foxp3 DTR mice; T.B. provided Foxp3 DTR mice; B.W. performed some gain- and loss-of-function experiments and supported IF and IHC quantifications; M.M. supported and supervised proteomic analyses; S.C.W. contributed to the writing of the manuscript; M.H.T. provided substantial contribution to conceptualization and discussion of the project and contributed to writing of the manuscript; C.D. conceptualized, designed, analyzed and interpreted data and wrote the manuscript.

## Funding

## Competing interests

As a scientist, M.H.T. participated in a scientific advisory board meeting of ERX Pharmaceuticals, Inc., Cambridge, MA, in 2019. He was a member of the Research Cluster Advisory Panel (ReCAP) of the Novo Nordisk Foundation between 2017 and 2019. He received funding for his research projects by Novo Nordisk (2016–2020) and Sanofi-Aventis (2012–2019). He consulted twice for Böhringer Ingelheim Pharma GmbH & Co. KG (2020 & 2021) and delivered a scientific lecture for Sanofi-Aventis Deutschland GmbH (2020) and for AstraZeneca GmbH (2024). As CEO and CSO of Helmholtz Munich, he is co-responsible for countless collaborations of the employees with a multitude of companies and institutions, worldwide. In this capacity, he discusses potential projects with and has signed/signs contracts for the centers institute(s) related to research collaborations worldwide, including but not limited to pharmaceutical corporations like Boehringer Ingelheim, Novo Nordisk, Roche Diagnostics, Arbormed, Eli Lilly, SCG Cell Therapy and others. As the CEO of Helmholtz Munich, he was/is further overall responsible for commercial technology transfer activities. M.H.T. confirms that to the best of his knowledge, none of the above funding sources or collaborations were involved in or had an influence on the preparation of this manuscript. The remaining authors declare no competing interests.

## Additional information

[1]Research Unit Type 1 Diabetes Immunology, Helmholtz Diabetes Center at Helmholtz Munich, Munich, Germany. [2]German Center for Diabetes Research (DZD), Munich, Germany. [3]Institute for Diabetes and Obesity, Helmholtz Diabetes Center at Helmholtz Munich and Division of Metabolic Diseases, Technische Universität München, Munich, Germany. [4]Department of Neurosurgery, Medical Faculty, Johannes Kepler University Linz, Linz, Austria. [5]Clinical Research Institute for Neurosciences, Johannes Kepler University Linz and Kepler University Hospital, Linz, Austria. [6]Neurosurgical Research, Department of Neurosurgery, University Hospital, Ludwig-Maximilians-University Munich, Munich, Germany. [7]Institute for Diabetes Research, Helmholtz Diabetes Center at Helmholtz Munich, 80939 Munich, and Klinikum rechts der Isar, Technische Universität München, Munich, Germany. [8]Department of Proteomics and Signal Transduction, Max Planck Institute of Biochemistry, Martinsried, Germany. [9]Institute for Molecular Medicine, Universitätsmedizin der Johannes-Gutenberg-Universität, Mainz, Germany. [10]Institute for Anatomy, Leipzig University, Leipzig, Germany. [11]Animal Physiology and Immunology, Technische Universität München, Freising-Weihenstephan, Germany. [12]Genomics Core Facility, EMBL European Molecular Biology Laboratory, Heidelberg, Germany. [13]Department of Endocrinology and Metabolism, Amsterdam University Medical Center, location AMC, University of Amsterdam, Amsterdam, The Netherlands. [14]Institute of Immunology, Johannes Gutenberg University Mainz, Mainz, Germany. [15]Metabolic Diseases Institute, Department of Psychiatry and Behavioral Neuroscience, University of Cincinnati, Cincinnati, OH, USA. [16]Department of Medicine 1, University of Erlangen-Nuremberg, Kussmaul Campus for Medical Research, Erlangen, Germany. [17]Division of Clinical Pharmacology, Department of Medicine IV, Ludwig-Maximilians-Universität München, Munich, Germany. ✉e-mail: matthias.tschoep@helmholtz-munich.de; carolin.daniel@helmholtz-munich.de

