## [Peer review file · Nature Communications]

Regulatory T cells in the mouse hypothalamus control immune activation and ameliorate metabolic impairments in high-calorie environments

Corresponding Author: Professor Carolin Daniel

Version 0:

Reviewer comments:

Reviewer #1

(Remarks to the Author)

The paper by becker et al. presents data on a highly topical issue. It is a fundamentally exciting paper that is methodologically elaborate in its implementation.

There are some weaknesses in the paper that need to be addressed.

Abstract: There should be a sentence about the functional role of the Tregs in the hypothalamus.

Introduction: Line 120: a sentence on how metabolic overload induces inflammation in metabolic tissues would be helpful. A short reference to CD4 cells would be helpful.

Results: Line 182-line 186: Here, some discussion is already anticipated.

What I miss in the paper is the analysis of other important expression profiles of classical inflammatory mediators in the hypothalamus associated with NFκB. This would make it a bit clearer to what extent the altered CD4 cell profile triggers inflammatory downstream processes that may also radiate to other regions of the brain.

An overview of the body weights of the animal groups would be helpful.

Discussion: In the discussion, I would like to see a stronger reference to other compartments, as has been studied in the approach in adipose tissue. In some places the authors do this, but stop where it goes into detail.

Are there any strategies to increase the number of Tregs in the CNS? This seems to have a protective effect on the immunological milieu in the hypothalamus. This should be discussed.

There is something missing in the discussion the limitation of individual methods

Further comments:

Did the experimental animals develop systemic inflammation?

Do the authors think that behavioral changes, such as sickness behavior, can be inferred from the findings?

Is there any data that a reduction of the Treg cell milieu also has a protective effect in adipose tissue? The authors should either use this as a comparison or introduce it as an impulse at the end of the discussion.

Reviewer #2

(Remarks to the Author)

This intriguing manuscript by Becker and colleagues examines the role of regulatory T cells (Tregs) in the hypothalamic immune response to the consumption of HFHS diet. The authors elegantly performed gain- and loss-of-function experiments to conclude that hypothalamic Tregs restrict diet-induced microglial activation and infiltrating macrophages. Collectively, these are novel findings and very significant to the field. However, the paper does not directly explore the ultimate question: what is the role of hypothalamic Tregs in the regulation of energy homeostasis? Since hypothalamic immune activation mediates obesity susceptibility, the presence or absence of hypothalamic Tregs should influence feeding and body weight regulation. For instance, have the authors considered to use the gain-of-function model in DIO mice? If their hypothesis is correct, the recovery of Tregs should be associated with metabolic benefits. The absence of such data limits enthusiasm.

In addition to my main concern stated above, the following comments should be addressed by the authors:

1. The presence of Tregs in the hypothalamus is extremely low. It has been recently estimated around 150 Tregs in the whole healthy adult mouse brain (Pasciuto et al, 2020). For that reason, the use of flow cytometry analyses for Tregs is totally reasonable. However, is this the best approach to assess hypothalamic microglial activation in response to HFHS diet? It seems that microglial inflammatory response is very confined to only some hypothalamic nuclei such as arcuate nucleus or paraventricular nucleus of the hypothalamus so therefore histological analysis of microglia in situ in these specific nuclei would be necessary as authors show in Fig S5K. Moreover, microglia are very sensitive to ex vivo artifacts due to tissue dissociation protocols. Also, how about measuring gene expression or protein levels of inflammatory cytokines in the hypothalamus?
2. The authors should clarify the inconsistency with the duration of diet interventions. For instance, they show reduced hypothalamic Tregs after 8 and 16 weeks of HFHS diet (Fig 20) but increased microglia activation only after 16 weeks (Fig 4 C) or 12 weeks in CX3CR1-GFP mice (Fig 4G). Why is the proteomic analysis performed in mice fed on HFHS diet for 36 weeks? Hypothalamic microglia activation occurs very quickly in response to HFHS diet even preceding body weight gain. Is this a unique obesity-induced hypothalamic immune activation instead of diet-induced hypothalamic immune activation?
3. What is the rationale of comparing ob/ob mice with WT-HFHS mice? Ob/ob mice don't have hypothalamic immune activation unless they are fed a HFHS diet (Gao et al, 2014).
4. In Fig S5K, is this a representative image of HFHS diet? What about the standard diet control? It's important to have both because based on their anatomical location, these MHCII+ cells seem to be meningeal myeloid cells (always there independently of diet) instead of parenchymal microglia.

Reviewer #3

(Remarks to the Author)

This paper by Becker and colleagues investigates the link between a high-calorie diet and immune activation within the hypothalamus including activation of resident macrophages and microglial cells and an increase in CD4 T effector memory T cells with a Th1-like profile. They demonstrate that this increased activation is associated with a reduction in the frequency of Tregs and by a set of gain and loss of function experiments, suggesting that changes in Tregs caused by the high-calorie diet is causal in allowing the immune activation. Overall, the studies are well conducted and, although there are some limitations, present a body of experimental data that on balance supports this hypothesis.

Some points for consideration:

Regarding the transcriptome analysis of hypothalamus-residing CD4+T cells:

Firstly, for the sake of transparency, it is important to show a representative flow cytometry plot of the cell sorting strategy to allow the reader to understand better the cells that were isolated (for both CD4+CD25-CD44^{low}+int T cells and CD4+CD25+Foxp3GFP+Tregs).

Secondly, given the continuum of expression of CD44, were the CD4+CD25-CD44^{low}+int cells isolated from HFHS and normal mice similar in phenotype of CD44 or, for example, is there an indication that the HFHS mice had a higher proportion of CD44^{lo} cells indicating a fundamental difference in the maturational status of the input cell populations~?

Thirdly, it is not entirely clear why a transcriptional analysis of the cell population expanded by the HFHS diet (presumably CD44^{hi}CD62L^{lo}) was not also performed.

The second most highly upregulated transcripts in CD4+CD25+Foxp3GFP+Tregs from HFHS-fed mice is *il7r*. The authors should comment on this observation given its proposed role in tissue Treg homeostasis.

Figure 5 should include an analysis of frequencies of CD80+, CD86+ and MHCII+ cells in macrophages from Balb/c mice.

The discussion should address the issues of timing in the proposed model. Activation of macrophages and microglial cells is demonstrated as early as 1-week post-HFHS diet but the reduction of Treg is first shown at 8 weeks and is relatively mild at this point. Although a causal role for Treg is suggested by the GOF experiments, the authors should discuss whether the HFHS diet is affecting Treg function prior to their loss from the hypothalamus.

Minor points:

The authors equate a CD62L low phenotype with an activated T cell phenotype. It may be more accurate to describe this as an 'effector memory' phenotype, as CD62L low staining does not strictly indicate recent activation.

Line 227-228: the increase in local CD4+T cell frequency after chronic exposure to an HFHS diet should be reported as not significant unless statistical analysis proves otherwise.

Line 239-241: the statement 'Activation of CD4+T cells upon HFHS diets was

confirmed by increased frequencies of CD4+CD62LlowCD44highTcells in the the hypothalamus (Figure 2K) and to a lesser extent also in the remaining parts of the brain (Figure 2L) should either be backed up by statistical analysis (presumably this analysis is available for all the animals shown in Figure 2 I+J) or removed.

The authors should comment on the increase in the frequency of Tregs observed in the mice on HFHS diet @48w (Figure 2M)

Line 310: the figure legend states the cells sorted were ed CD4+CD25-CD44low+int T cells. This detail should be added in the results section.

Details should be given on how the placement of gates was determined for all flow cytometry studies. This is particularly important for the analysis of markers no showing distinct populations positive for the maker (e.g. MHC-class II and co-stimulatory marker expression on myeloid cells).

Line 438-439: The authors suggest that post Treg depletion, enhanced immune activation is lower in other parts of the brain compared to the hypothalamus but I do not believe the data supports this.

Line 579: 'frequent' should be changed to 'frequency'

Version 1:

Reviewer comments:

Reviewer #1

(Remarks to the Author)

All my recommendations are included. I recommend to publish the manuscript.

Reviewer #2

(Remarks to the Author)

The authors carefully addressed all my concerns and this manuscript is now improved and suitable for publication.

Reviewer #4

(Remarks to the Author)

The investigators addressed well enough all but

- The reviewer's request for transcriptional data on CD44hi CD62lo (activated) non-Treg cells is a valid one because tissue Tregs are generally in an activated, not naïve, state. The authors did not adequately address this point.
- The reviewer's comment on timing is valid but the authors' answer is unconvincing because
 - o GOF experiments only tell what CAN happen, not what actually does happen
 - o Upregulation of *Tnfrsf14* is a picked cherry of unknown significance
 - o Treg suppression assays on lymphoid organ Tregs are not meaningful in this context.
- Concerning "frequent" vs "frequency": the reviewer is correct and the authors are incorrect. If others have used "frequent" in this context, they must be non-native English speakers.

Version 2:

Reviewer comments:

Reviewer #4

(Remarks to the Author)

This revised version should be accepted for publication.

Point-by-point response

Nature Communications manuscript NCOMMS-22-21119

REVIEWER COMMENTS

Reviewer #1 (Remarks to the Author):

The paper by Becker et al. presents data on a highly topical issue. It is a fundamentally exciting paper that is methodologically elaborate in its implementation. There are some weaknesses in the paper that need to be addressed.

Response: We thank the reviewer for this enthusiastic feedback on the relevance of our findings. We also thank him/her for highlighting some weaknesses which we addressed during the revision process. This constructive advice supported us in further strengthening the conclusions of the manuscript.

Abstract:

There should be a sentence about the functional role of the Tregs in the hypothalamus.

Response: We thank the reviewer for this advice and have added such a sentence within the abstract. For further details, please refer to the manuscript.

Introduction:

Line 120: a sentence on how metabolic overload induces inflammation in metabolic tissues would be helpful.

Response: We thank the reviewer for this advice and have added such a sentence within the abstract. For further details, please refer to the manuscript.

A short reference to CD4 cells would be helpful.

Response: We thank the reviewer for this comment and have included such a sentence within the abstract. For further details, please refer to the manuscript.

Results:

Line 182-line 186: Here, some discussion is already anticipated. What I miss in the paper is the analysis of other important expression profiles of classical inflammatory mediators in the hypothalamus associated with NFκB. This would make it a bit clearer to what extent the altered CD4 cell profile triggers inflammatory downstream processes that may also radiate to other regions of the brain.

Response: We thank the reviewer for this advice. We have now added novel experimental data to the manuscript to address this question. Specifically, we have performed qPCR analyses from hypothalamic tissue in response to hypercaloric challenge. Here, as a NF-κB target gene we identified significantly increased expression of *Ccl5* (**Figure S2c**). Trends towards higher expression of *Ccl5* and *cRel* and *RelB* were also observed in the remaining brain (**Figure S2d**). In line with this observation and as detailed in the manuscript, RNA-seq data of hypothalamic CD4⁺T cells following a high-caloric challenge likewise showed upregulated *Ccl5* as well as reduced expression of *Ccr7* and *S1pr1*, supporting their recruitment to inflammatory tissues as well as their retention at these sites. In an additional set of novel experiments, we confirmed involvement of NFκB signaling upon HFHS diet exposure using isolated cellular populations from the brain. Specifically, we purified ACSA2⁺ astrocytes from the hypothalamus of HFHS diet-exposed animals. Astrocytes from these mice revealed a significant increase in *Ikkb* expression when compared to standard diet-fed animals. These novel findings are now presented in **Figure S2d**. We found no significant changes in *Ikkb* and related *Tnfa* expression in astrocytes in remaining parts of the brain (**Figure S2d**). In addition, we isolated microglia from such mice and observed trends towards increased *Ikkb* and *Tnfa* expression in both the hypothalamus and remaining brain (**Figure S2d**). Likewise, similar trends for *Ikkb* and *Tnfa* abundance were seen in isolated NeuN⁺ neurons in the remaining brain (**Figure S2d**). In light of the critical relevance of neuronal-immune crosstalk upon hypercaloric feeding, we validated the expression of NFκB related signaling in purified NeuN⁺ neurons in an independent setting of HFHS-diet exposed Balb/c mice. Accordingly, in NeuN⁺ neurons from animals fed HFHS-diet we observed a significantly increased abundance of *Ccl5*, *cRel*, *Relb*, *Ikkb* and *Tnfa* (**Figure S2e**). For further details, please refer to the results section of the manuscript.

An overview of the body weights of the animal groups would be helpful.

Response: The respective body weights have been included. For further details, please refer to **Figures 7c, S1e, S1f** and **S2f**.

Discussion: In the discussion, I would like to see a stronger reference to other compartments, as has been studied in the approach in adipose tissue. In some places the authors do this, but stop where it goes into detail.

Response: We thank the reviewer for this helpful suggestion. We have now referred to other compartments in the discussion section of the manuscript. In addition, we have also included novel data obtained during the revision of the manuscript. Specifically, we have now studied individual cellular populations in the brain including astrocytes, microglia and neurons upon hypercaloric feeding and focused on NFkB-related signaling pathways. Moreover, following Treg targeting we assessed inflammatory and metabolic characteristics in the adipose tissue (see **Figure S8d-e**) and investigated glucose tolerance among others (see **Figure 7f-g, Figure 8i-j, Figure 9g**). For further details, please refer to the results and the discussion section of the manuscript.

Are there any strategies to increase the number of Tregs in the CNS? This seems to have a protective effect on the immunological milieu in the hypothalamus. This should be discussed.

Response: We thank the reviewer for this important question. We have approached the question of increasing Tregs in the CNS experimentally and have included novel Treg-gain-of function studies in the manuscript. These novel results are now presented in the results section of the manuscript. Specifically, for specific Treg expansion, we employed established procedures of anti-IL2/IL2 antibody complexes¹. Importantly, Treg expansion resulted in significantly reduced expression of NFkB target gene *Ccl5* in the hypothalamus (**Figure 7d**). Furthermore, Treg expansion significantly reduced NFkB-related genes including *RelB* and *Tnfa* in the subcutaneous and visceral adipose tissue (**Figure S8d-e**). Of note, specific Treg expansion significantly ameliorated the negative impact of hypercaloric feeding as seen from reduced body weight, fat mass and improved glucose tolerance. These novel results are now presented in **Figure 7** and **Figure S8**. Possible additional strategies to increase local

Tregs include an IL-33-mediated expansion of ST2⁺ Tregs^{2, 3} or delivery of IL-2 into the CNS/astrocytes⁴. Future studies will be required to further advance possibilities of Treg targeting in a niche- and context-specific manner.

There is something missing in the discussion the limitation of individual methods

Response: In line with the advice from this reviewer, we have now detailed the limitation of individual methods following the discussion section of the manuscript. For further details, please refer to the manuscript.

Further comments:

Did the experimental animals develop systemic inflammation?

Response: In line with previous studies using HFHS diets in mice, upon continued exposure to hypercaloric feeding the animals developed low-grade systemic inflammation. To verify this mild inflammation, we now assessed pro-inflammatory cytokine production in CD4⁺T cells of lymph nodes from HFHS diet-exposed mice. These novel data have now been included in the manuscript. Accordingly, cytokine stainings showed low levels of IL-17A production in CD4⁺T cells upon HFHS diet, but still a significant increase in comparison to animals fed a standard diet (**Figure S2a**). We did not observe any significant changes in IFN γ production in CD4⁺T cells purified from lymph nodes of mice exposed to HFHS diet vs. standard diet (**Figure S2a**). Similarly, we now performed novel experiments and assessed the suppressive potential of Tregs purified from HFHS diet vs. standard diet fed mice and found no significant difference in the ability of Tregs to inhibit the proliferation of conventional T cells. These novel results have now been included in **Figure S1j**.

Do the authors think that behavioral changes, such as sickness behavior, can be inferred from the findings?

[REDACTED]

[REDACTED]

[REDACTED]

Is there any data that a reduction of the Treg cell milieu also has a protective effect in adipose tissue? The authors should either use this as a comparison or introduce it as an impulse at the end of the discussion.

Response: We thank the reviewer for this interesting comment. Yes - work from Bapat et al. ⁶ suggest that in contrast to mice with obesity associated insulin resistance (IR) animals with age-associated IR benefit from a reduction in adipose tissue Tregs upon depletion of these cells using anti-ST2 antibodies. Albeit, anti-ST2 antibodies do not only deplete Tregs in fat but also other metabolic tissues, these data are in line with the concept that distinct immune cell populations within adipose tissue underlie aging- vs. obesity-related metabolic aberrations. In accordance with that assumption, it will be of interest in future studies to assess the implications of Tregs in the CNS in modulating age-associated immune-metabolic alterations in the adipose tissue. Following the advice of this reviewer, we have now added a paragraph in relation to these data as an impulse at the end of the discussion – for further details please refer to the corresponding discussion section of the manuscript.

Reviewer #2 (Remarks to the Author):

This intriguing manuscript by Becker and colleagues examines the role of regulatory T cells (Tregs) in the hypothalamic immune response to the consumption of HFHS diet. The authors elegantly performed gain- and loss-of-function experiments to conclude that hypothalamic Tregs restrict diet-induced microglial activation and infiltrating macrophages. Collectively, these are novel findings and very significant to the field.

Response: We thank this reviewer for the positive feedback on the relevance and importance of our findings presented in this manuscript.

However, the paper does not directly explore the ultimate question: what is the role of hypothalamic Tregs in the regulation of energy homeostasis? Since hypothalamic immune activation mediates obesity susceptibility, the presence or absence of hypothalamic Tregs should influence feeding and body weight regulation.

Response: We thank the reviewer for this comment and important advice. Following the suggestion of this reviewer, we have now performed a series of novel studies to assess anti-inflammatory and metabolic benefits in response to Treg-gain-of function. Specifically, we used specific *in vivo* Treg expansion with anti-IL2/IL2 antibody complexes followed by exposure to HFHS diet. Importantly, local Treg expansion

resulted in a significant decline in HFHS-diet induced inflammatory responses such as *Ccl5* expression in the hypothalamus. Likewise, *Ccl5* and related NF-kB targets revealed trends toward a reduction in remaining parts of the brain of mice receiving specific Treg expansion together with reductions in NFkB-related genes as *RelB*, *cRel* and *Tnfa* (**Figure 7** and **Figures S8b-e**). Importantly, to link reduced local inflammation upon Treg expansion with systemic metabolic benefits we studied body weight and fat masses. Importantly, specific Treg expansion fostered a significant reduction in body weight and fat mass gain in response to HFHS diet (**see Figure 7c**). In addition, we performed a series of glucose tolerance tests to define further the role of Treg expansion on systemic energy metabolism. Here, Treg-gain-of-function supported a significant improved glucose tolerance following 4 vs. 8 weeks of HFHS diet (**Figures 7f and 7g**).

To validate the relevance of Treg gain-of-function in an independent experimental setting we used a direct cellular approach and performed peripheral transfers (*i.v.*) of *Foxp3*⁺*GFP*⁺Tregs into C57Bl/6J-recipient mice followed by exposure to the HFHS diet. In line with the reduced hypothalamic immune activation as detailed in the results section of the manuscript, Treg-gain-of-function using Treg transfers supported an improved glucose tolerance ($p=0.0531$) after 4 and 8 weeks ($p<0.0001$) of the HFHS diet, again underlining a direct metabolic benefit induced by local Tregs.

Of note, in another independent strategy we employed direct *i.c.* transfer of Tregs into the hypothalamus. These Treg transfers into the hypothalamus resulted in reduced local immune activation and supported a clear trend towards improved glucose tolerance ($p=0.0643$, **Figure 9g**) accompanied by trends towards reduced inflammatory NFkB-related genes in the adipose tissue (**Figure S9e**).

These findings are in accordance with the other Treg-gain-of-function approaches as detailed here and indicate that Tregs in the CNS lower diet-induced immune activation and improve systemic metabolism. For further details, please refer to the results and discussion section of the manuscript.

For instance, have the authors considered to use the gain-of-function model in DIO mice? If their hypothesis is correct, the recovery of Tregs should be associated with metabolic benefits. The absence of such data limits enthusiasm.

Response: We thank the reviewer for this helpful suggestion. As detailed in our response to the comment above, we have now performed a series of novel experimental studies to show that Treg-gain-of-function strategies do support systemic metabolic benefits including reduced body weight gain, fat mass and improved glucose tolerance. These novel findings are now presented in **Figures 7, 8 and 9** and the corresponding supplemental **Figures S8 and S9**.

In addition to my main concern stated above, the following comments should be addressed by the authors:

1. The presence of Tregs in the hypothalamus is extremely low. It has been recently estimated around 150 Tregs in the whole healthy adult mouse brain (Pasciuto et al, 2020). For that reason, the use of flow cytometry analyses for Tregs is totally reasonable. However, is this the best approach to assess hypothalamic microglial activation in response to HFHS diet? It seems that microglial inflammatory response is very confined to only some hypothalamic nuclei such as arcuate nucleus or paraventricular nucleus of the hypothalamus so therefore histological analysis of microglia in situ in these specific nuclei would be necessary as authors show in Fig S5K. Moreover, microglia are very sensitive to ex vivo artifacts due to tissue dissociation protocols.

Response: We thank the reviewer for this important comment and would like to highlight that we used no enzymatic digestion protocol for isolation and corresponding analyses of T cells or microglia via flow cytometry. Instead and to avoid the possibility of ex vivo artifacts due to dissociation protocols, we employed gentle manual dissociation using Dounce homogenizers. Furthermore, the high quality of the flow cytometric analyses was ensured by using appropriate gating strategies and the use of a set of exclusion markers and viability stains as indicated in the corresponding figures and FACS plots shown in the Figures.

Importantly, these experimental studies using flow cytometry have also been validated using independent experimental approaches e.g. by immunofluorescence. In **Figure S6k**, we have now included the co-staining of MHCII and Iba1 in the median eminence of standard diet vs HFHS diet-fed mice. On those microscopic images, it is clearly visible that upon HFHS diet feeding microglia undergo reactive gliosis, which can be

observed by a change in phenotype and Iba1 staining intensity. The MHCII staining which is mainly limited to meningeal myeloid cells in the standard diet, spreads into the brain parenchyma when mice are exposed to the HFHS diet (**Figure S6k**).

Also, how about measuring gene expression or protein levels of inflammatory cytokines in the hypothalamus?

Response: We thank the reviewer for this helpful comment. We have now performed a series of novel studies to assess inflammatory markers in the hypothalamus, brain and adipose tissue in further detail. In line with the suggestion of Reviewer 1, we also focused on NFkB-related signaling and target genes and have identified that hypercaloric challenges promotes a significant increase in the NFkB-related genes such as *Ccl5* in the hypothalamus. For further details, please refer to the results section of the manuscript and **Figures 7d**, **Figure S2d-e** and **Figure S8b-c**.

2. The authors should clarify the inconsistency with the duration of diet interventions. For instance, they show reduced hypothalamic Tregs after 8 and 16 weeks of HFHS diet (Fig 20) but increased microglia activation only after 16 weeks (Fig 4 C) or 12 weeks in CX3CR1-GFP mice (Fig 4G). Why is the proteomic analysis performed in mice fed on HFHS diet for 36 weeks? Hypothalamic microglia activation occurs very quickly in response to HFHS diet even preceding body weight gain. Is this a unique obesity-induced hypothalamic immune activation instead of diet-induced hypothalamic immune activation?

Response: We thank the reviewer for this important question. We would like to highlight that one of our focus within the framework of this manuscript was to define the relevance and contribution of adaptive immune cells – specifically CD4⁺T cells – in the development of local immune activation in the hypothalamus. Specifically, we aimed to dissect the kinetics of changes of the adaptive arm of the immune system induced in response to hypercaloric challenges. This contribution not only includes numerical but mainly phenotypic alterations of CD4⁺T cells and Foxp3⁺Tregs in response to hypercaloric feeding. We agree that microglia activation occurs quickly in response to exposure to a HFHS diet and this has been shown robustly by numerous publications. However, the analyses we performed were focusing in addressing the

question, if, when and how protective immune mechanisms of regulatory Foxp3⁺Tregs can impinge on microglia cells. Therefore, it was required to study various durations of diet interventions. The corresponding body weight curves were included in the revised version of the manuscript (e.g. **Figure S1e-f, S2f, Figure 7c**).

3. What is the rationale of comparing ob/ob mice with WT-HFHS mice? Ob/ob mice don't have hypothalamic immune activation unless they are fed a HFHS diet (Gao et al, 2014).

Response: We thank the reviewer for this important question. Here, we would like to highlight that the hypothalamic immune activation that is described in numerous robust publications always refers to the innate arm of the immune system. To the best of our knowledge, there are no data published so far that analyze the adaptive arm of the immune system i.e. CD4⁺ T cells or Foxp3⁺ regulatory T cells in hypothalamus or other parts of the brain of ob/ob mice. Therefore, the comparison of a genetically obese model i.e. the ob/ob mouse with wildtype mice fed a HFHS diet is valid and relevant when focusing on CD4⁺T cells and Foxp3⁺Tregs. We have now included new experimental data in **Figure S2f-g**. Specifically, we show that ob/ob mice that are fed a HFHS diet have significantly less Foxp3⁺Tregs. Nevertheless, we have now included additional analyses of CD4⁺ T cells isolated from ob/ob mice on a SD vs ob/ob mice fed a HFHS diet that reveal a significant reduction in Foxp3⁺ Treg frequencies in brains of ob/ob mice upon hypercaloric challenge (**Figure S7g**).

4. In Fig S5K, is this a representative image of HFHS diet? What about the standard diet control? It's important to have both because based on their anatomical location, these MHCII⁺ cells seem to be meningeal myeloid cells (always there independently of diet) instead of parenchymal microglia.

Response: We thank the reviewer for this question. We have now additionally included the corresponding staining of the median eminence of a SD-fed animal in **Figure S6k**.

We agree that a large proportion of MHCII expressing cells might be meningeal myeloid cells. However, under HFHSD conditions also parenchymal cells start expressing MHCII as shown in this representative figure. This becomes even clearer

when the SD staining is compared to the HFHSD: a distinct sign of reactive microgliosis/microglial activation together with the presence of the MHCII staining.

Reviewer #3 (Remarks to the Author):

This paper by Becker and colleagues investigates the link between a high-calorie diet and immune activation within the hypothalamus including activation of resident macrophages and microglial cells and an increase in CD4 T effector memory T cells with a Th1-like profile. They demonstrate that this increased activation is associated with a reduction in the frequency of Tregs and by a set of gain and loss of function experiments, suggesting that changes in Tregs caused by the high-calorie diet is causal in allowing the immune activation. Overall, the studies are well conducted and, although there are some limitations, present a body of experimental data that on balance supports this hypothesis.

Response: We thank the reviewer for the positive feedback concerning the performance, relevance and importance of the proposed studies.

Some points for consideration:

Regarding the transcriptome analysis of hypothalamus-residing CD4+T cells:

Firstly, for the sake of transparency, it is important to show a representative flow cytometry plot of the cell sorting strategy to allow the reader to understand better the cells that were isolated (for both CD4+CD25-CD44^{low+int} T cells and CD4+CD25+Foxp3GFP+Tregs).

Response: We thank the reviewer for this comment. A representative FACS plot showing the cell sorting strategy was now included in **Figure S2a**.

Secondly, given the continuum of expression of CD44, were the CD4+CD25-CD44^{low+int} cells isolated from HFHS and normal mice similar in phenotype of CD44 or, for example, is there an indication that the HFHS mice had a higher proportion of CD44^{lo} cells indicating a fundamental difference in the maturational status of the input cell populations~?

Response: We thank the reviewer for this important comment. The gating used for sorting is now included in **Figure S3a**. Furthermore, we have carefully analyzed the data obtained with regard to differences in input cell populations. As shown in the new **Figure S2b**, the input cell populations did not differ with respect to their abundance of CD44 as analyzed by CD44 mean fluorescence intensity (MFI) – per cell. Hence, sorted cell populations used as input for sequencing were not different in their maturational status with regard to the markers employed for sorting but uniform naïve T cells.

Thirdly, it is not entirely clear why a transcriptional analysis of the cell population expanded by the HFHS diet (presumably CD44^{hi}CD62L^{lo}) was not also performed.

Response: We thank the reviewer for this relevant question. As mentioned by the reviewer in the previous question, we aimed to avoid diluting possible effects caused by the HFHS diet feeding by analyzing poorly-defined cell populations such as total CD4⁺T cells. Such total CD4⁺T cells from SD vs HFHS diet-fed animals would consist of undefined (and phenotypically different) input cell populations, thereby diluting any potentially interesting differences (i.e. genes that might be differently regulated in naïve T cells vs. activated T cells etc. as response to hypercaloric feeding). Given our strong interest in pro-tolerogenic (anti-inflammatory) environments and given the fact that Foxp3⁺ regulatory T cells can be induced efficiently from naïve T cells *in vivo*, we decided that particularly the naïve T cell population is of major interest for our analyses.

The second most highly upregulated transcripts in CD4⁺CD25⁺Foxp3⁺GFP⁺Tregs from HFHS-fed mice is *il7r*. The authors should comment on this observation given its proposed role in tissue Treg homeostasis.

Response: We thank the reviewer for this helpful comment. We have now outlined the respective literature concerning *il7r* expression in Tregs from HFHS-exposed animals. Specifically, we indicate previous work from Iris Gratz et al. ⁷ showing that memory Tregs do require IL-7 for their maintenance in the skin. Accordingly, this concept could suggest that the upregulation of *il7r* upon exposure to HFHS diet may serve rather as a compensatory mechanism to safeguard their maintenance under conditions of hypercaloric challenge. In addition, we also discuss recent findings

suggesting that the upregulation of IL-7R α expression strongly interfered with IL-2 receptor signaling in Treg cells accompanied by blunted downstream Stat5 phosphorylation⁸. These results could indicate a reduced proliferative potential of Tregs in response to HFHS diet based on high levels of IL7R expression.

Figure 5 should include an analysis of frequencies of CD80+, CD86+ and MHCII+ cells in macrophages from Balb/c mice.

Response: We thank the reviewer for this comment. We carefully checked the data presented in the manuscript. However, to the best of our knowledge the data the reviewer requested are already presented in the manuscript. For further details, please refer to the Figure indicated below which is now shown as CD80/CD86/MHCII (% of macrophages) in **Figure 5g-i** of the re-submitted manuscript.

Figure 5

The discussion should address the issues of timing in the proposed model.

Activation of macrophages and microglial cells is demonstrated as early as 1-week post-HFHS diet but the reduction of Treg is first shown at 8 weeks and is relatively mild at this point. Although a causal role for Treg is suggested by the GOF

experiments, the authors should discuss whether the HFHS diet is affecting Treg function prior to their loss from the hypothalamus.

Response: We thank the reviewer for this comment. To further emphasize the role of Tregs in mitigating HFHS-diet induced inflammatory and metabolic aberrations, we have now performed a series of novel Treg gain-of-function studies. Concerning functional alterations in Tregs, we have seen upregulation of *Tnfrsf14* expression in Tregs from HFHS diet-exposed mice, which might indicate impaired suppressive function of Tregs. Given the limited numbers of Tregs within the hypothalamus, it is not feasible experimentally to directly assess their cellular functionality through Treg suppression assays. However, we have performed such Treg suppression assays using Tregs isolated from lymph nodes of either from mice exposed to HFHS diet or standard diet. These functional studies did not reveal any significant impairments in Treg function in mice exposed to the HFHS diet (**Figure S1j**). Nevertheless, our data indicate that the deteriorating effects of hypercaloric feeding on Tregs is a local effect as observed specifically in the hypothalamus.

Minor points:

The authors equate a CD62L low phenotype with an activated T cell phenotype. It may be more accurate to describe this as an 'effector memory' phenotype, as CD62L low staining does not strictly indicate recent activation.

Response: We thank the reviewer for this suggestion. We have rephrased this sentence accordingly.

Line 227-228: the increase in local CD4+T cell frequency after chronic exposure to an HFHS diet should be reported as not significant unless statistical analysis proves otherwise.

Response: Thanks for this important comment. We agree and have rephrased this sentence accordingly.

Line 239-241: the statement 'Activation of CD4+T cells upon HFHS diets was confirmed by increased frequencies of CD4+CD62LlowCD44highTcells in the the hypothalamus (Figure 2K) and to a lesser extent also in the remaining parts of

the brain (Figure 2L) should either be backed up by statistical analysis (presumably this analysis is available for all the animals shown in Figure 2 I+J) or removed.

Response: We thank the reviewer for this comment. We have accordingly complemented the data now presented in **Figure 2m**.

The authors should comment on the increase in the frequency of Tregs observed in the mice on HFHS diet @48w (Figure 2M)

Response: We thank the reviewer for this comment. We have now also eluded on this in the results section of the manuscript. Specifically, we have indicated that: Long-term exposure to such hypercaloric challenges promoted a significant increase of Tregs in LNs, which could result among others from inflammatory expansion or retention.

Line 310: the figure legend states the cells sorted were ed CD4+CD25-CD44low+int T cells. This detail should be added in the results section.

Response: We followed the reviewer's advice and have included this information accordingly in the results section.

Details should be given on how the placement of gates was determined for all flow cytometry studies. This is particularly important for the analysis of markers no showing distinct populations positive for the maker (e.g. MHC-class II and co-stimulatory marker expression on myeloid cells).

Response: We thank the reviewer for this comment. We have rephrased the corresponding methods section carefully to highlight that fluorescence minus one (FMO) controls were used to set gates. For some analyses the cell populations expressing high amounts of a certain marker (such as MHCII) were of interest. Therefore, for these analyses, the gates were not placed according to being positive or negative, but to reflect cell populations that have very high amounts of the corresponding protein.

Line 438-439: The authors suggest that post Treg depletion, enhanced immune activation is lower in other parts of the brain compared to the hypothalamus but I do not believe the data supports this.

Response: We thank the reviewer for this comment. We agree and have rephrased that sentence carefully. For further details, please refer to the results section of the manuscript.

Line 579: 'frequent' should be changed to 'frequency'

Response: In this regard, we would like to friendly disagree with this reviewer. We did not see a reason to change the wording 'low frequent' to 'frequency' given that the term 'low frequent' is commonly used.

References

1. Webster KE, *et al.* In vivo expansion of T reg cells with IL-2-mAb complexes: induction of resistance to EAE and long-term acceptance of islet allografts without immunosuppression. *J Exp Med* **206**, 751-760 (2009).
2. Ito M, *et al.* Brain regulatory T cells suppress astrogliosis and potentiate neurological recovery. *Nature* **565**, 246-250 (2019).
3. Han JM, Wu D, Denroche HC, Yao Y, Verchere CB, Levings MK. IL-33 Reverses an Obesity-Induced Deficit in Visceral Adipose Tissue ST2+ T Regulatory Cells and Ameliorates Adipose Tissue Inflammation and Insulin Resistance. *J Immunol* **194**, 4777-4783 (2015).
4. Yshii L, *et al.* Astrocyte-targeted gene delivery of interleukin 2 specifically increases brain-resident regulatory T cell numbers and protects against pathological neuroinflammation. *Nature Immunology*, (2022).
5. Ilanges A, Shiao R, Shaked J, Luo JD, Yu X, Friedman JM. Brainstem ADCYAP1(+) neurons control multiple aspects of sickness behaviour. *Nature* **609**, 761-771 (2022).
6. Bapat SP, *et al.* Depletion of fat-resident Treg cells prevents age-associated insulin resistance. *Nature* **528**, 137-141 (2015).
7. Gratz IK, *et al.* Cutting Edge: memory regulatory t cells require IL-7 and not IL-2 for their maintenance in peripheral tissues. *J Immunol* **190**, 4483-4487 (2013).
8. Waickman AT, *et al.* The Cytokine Receptor IL-7Ralpha Impairs IL-2 Receptor Signaling and Constrains the In Vitro Differentiation of Foxp3(+) Treg Cells. *iScience* **23**, 101421 (2020).

Point-by-point response

Revision #2

Nature Communications manuscript NCOMMS-22-21119

REVIEWER COMMENTS

Reviewer #1 (Remarks to the Author):

All my recommendations are included. I recommend to publish the manuscript.

Response: We would like to thank the reviewer for his/her positive feedback and recommendation to publish the manuscript.

Reviewer #2 (Remarks to the Author):

The authors carefully addressed all my concerns and this manuscript is now improved and suitable for publication.

Response: We would like to thank the reviewer for his/her positive feedback and recommendation to publish the manuscript.

Since reviewer #3 was **unavailable** to evaluate the revised version of the manuscript, reviewer #4 stepped in to provide feedback.

Reviewer #4 (Remarks to the Author): commenting on the answers to the previous concerns raised by Reviewer #3

Original comment of Reviewer 3: Thirdly, it is not entirely clear why a transcriptional analysis of the cell population expanded by the HFHS diet (presumably CD44^{hi}CD62L^{lo}) was not also performed.

Feedback from Reviewer 4:

The investigators addressed well enough all but

- The reviewer's request for transcriptional data on CD44^{hi} CD62L^{lo} (activated) non-Treg cells is a valid one because tissue Tregs are generally in an activated, not naïve, state. The authors did not adequately address this point.

Response to feedback from Reviewer 4: In line with the request of reviewer #4 (speaking for reviewer #3), we have now performed a series of novel experiments including a newly-executed RNA-seq experiment. Specifically, as requested by reviewer 4, we used FACS-sorted T cells (CD44^{hi} CD62L^{low}, hereafter referred to as **activated conventional T cells**) from brains and hypothalami of mice that were exposed to either a control diet or a high fat high sugar (HFHS) diet and subjected these cells as requested to RNA-seq analysis.

For this experiment, we obtained the Foxp3^{GFP} reporter mice from The Jackson Laboratories and subjected them to *ad libitum* feeding of the corresponding diets for 16-18 weeks, following

the same protocol used for Tregs and naïve T cells in the original version of the manuscript. To obtain sufficient T cell numbers for the RNA-seq analysis, we pooled brains or hypothalami from five mice per sample. The cells were stained for flow cytometry and sorted for purity using a BD Aria III cell sorter. A representative gating strategy for sorting these activated conventional T cells is shown for the convenience of reviewing in **Reviewer Figure 1** below and included in the manuscript in **Figure S4a**.

Reviewer Figure 1: Representative gating strategy for the FACS sort of activated conventional T cells ($CD44^{hi} CD62L^{low}$) from *Foxp3 GFP* reporter mice. These plots were now also included in Figure S4a. As indicated in the manuscript, the full gating strategy is: $live CD4^{+} CD25^{-} Foxp3GFP^{-} CD44^{hi} CD62L^{lo}$.

The obtained samples were processed using the SMART-Seq mRNA LP Kit (Takara) and the bioinformatics pipeline as before.

There were two main results obtained from this requested sequencing experiment:

1. The most prominently upregulated genes in response to HFHS diet were related to cell cycle. This was already expected by us and by Reviewer #3 since this cell population expands upon exposure to HFHSD. The results of this gene set enrichment analysis (GSEA) are included now in **Figure S4h** and shown here below in **Reviewer Figure 2** for your convenience.
2. The key phenotype/signature of activated conventional T cells are only mildly affected when mice are exposed to a HFHS diet. Our data show that the expression of well-established genes related to effector function, migration, or immune responses are not significantly changed when comparing FACS-sorted activated conventional T cells from mice exposed to different diets. This lack of significant variation is not unexpected, given that these effector cells have largely undergone terminal differentiation and we were FACS-sorting the very specific cell population ($CD4^{+} CD25^{-} Foxp3GFP^{-} CD44^{hi} CD62L^{lo}$) from both diet regimens. This holds true for activated conventional T cells sorted from hypothalami, as well as from brains of these mice. The data were included in **Figure S4b-g**. These novel results are presented/discussed in the results section of the manuscript on pages 15-16.

Reviewer Figure 2

Reviewer Figure 2: *a*) Gene set enrichment analysis (GSEA) comparing activated conventional T cells (Tconv) from brains of mice exposed to HFHS diet vs standard diet (SD). Depicted are pathways significantly upregulated in the cells isolated from HFHSD-fed mice (as indicated by the positive value of the normalized enrichment score). This figure was included in the revised manuscript in **Figure S4h**. *b*) Volcano plot in which genes of the leading edge on cell cycle terms are highlighted in red for transparency. This plot shows the data for activated conventional T cells isolated from hypothalami of HFHS diet vs. SD mice.

At this point we would like to emphasize that we conducted various careful analyses to ensure that the results presented here are not due to insufficient sequencing depth, experimental errors or problems in our bioinformatics pipeline. To demonstrate this here and in the manuscript, we plotted a direct comparison of activated conventional T cells with Tregs to show that, when comparing these two different T cell populations, the expected genes are detected. The corresponding volcano plot is depicted in **Reviewer Figure 3** and was included in the manuscript in **Figure S4f-g**.

Reviewer Figure 3

Reviewer Figure 3: Volcano plot showing the differentially expressed genes (DEG) of Tregs vs. activated conventional T cells (Tconv) that were isolated from brains of mice on HFHSD.

*A positive Log₂ fold change indicates genes that are upregulated in Tregs, while a negative Log₂ fold change refers to genes upregulated in Tconv. Dashed lines: cutoff of an adjusted p value <0.05 and Log₂ fold change >2 or < -2. Genes that are well-established for Tregs are highlighted in blue, genes expressed in activated conventional T cells are highlighted in yellow. The corresponding figure is included in the manuscript in **Figure S4g-h**.*

We would like to describe some of these well-established genes here: Tregs are characterized by their main transcription factor *Foxp3*, high amounts of the high-affinity alpha subunit of the Interleukin 2 receptor CD25 (encoded by *Il2ra*), *Irf4* (which encodes Eos), and *Ctla4*, which acts as an immune checkpoint molecule. These are well-established molecules which are involved in maintaining the suppressive function of Tregs. More of these are highlighted in **Reviewer Figure 3** in blue.

Interestingly, within this new dataset and this comparison, we were also able to detect *Areg*, which encodes for Amphiregulin. Amphiregulin has recently garnered increasing attention in laboratories investigating the role of tissue-residing Tregs. Within tissues, amphiregulin plays a critical role in promoting tissue repair and regeneration in the context of the resolution of inflammation. This has been demonstrated in various studies, including those ^{1,2,3,4}. In a study by Ito *et al.* ⁴, *Areg* expression by brain-residing Tregs was required to suppress excess astrogliosis, neurological deficits and neurotoxic gene expression in mouse models of ischemic brain injury. Another marker expressed by tissue Tregs, the receptor for IL-33, ST2 (which is encoded by *Il1rl1*) was significantly upregulated by Tregs in the brains of these mice.

On the side of the activated conventional T cells, we would like to highlight just a few genes that were significantly upregulated in this dataset: *Il2*, *Il4*, *Il17a* are all cytokines that are known to be released by activated conventional T cells to guide immune reactions and fulfill effector functions. Furthermore, these cells show higher expression of *Eomes*, encoding Eomesdesmin, which is known for its function in Th1 immune responses, together with *Ifng*, a major cytokine related to Th1 immunity.

These data were included in the revised manuscript in **Figure S4f-g** and presented in the results section of the manuscript (page 15 and the following).

To conclude, the data that were added in this new round of revision of the manuscript show that activated conventional T cells robustly proliferate upon exposure to HFHS diet, while the hypercaloric diet does not result in major significant changes in the key phenotype/signature of activated conventional T cell differentiation *per se*.

Original comment by reviewer 3: The discussion should address the issues of timing in the proposed model. Activation of macrophages and microglial cells is demonstrated as early as 1-week post-HFHS diet but the reduction of Treg is first shown at 8 weeks and is relatively mild at this point. Although a causal role for Treg is suggested by the GOF experiments, the authors should discuss whether the HFHS diet is affecting Treg function prior to their loss from the hypothalamus.

Feedback from reviewer 4 on the comment by previous reviewer 3:

- The reviewer's comment on timing is valid but the authors' answer is unconvincing because
 - o GOF experiments only tell what CAN happen, not what actually does happen
 - o Upregulation of *Tnfrsf14* is a picked cherry of unknown significance
 - o Treg suppression assays on lymphoid organ Tregs are not meaningful in this context.

Response to reviewer 3/4: We thank reviewers 3 and 4 for their valuable input. We have followed these recommendations and now included a paragraph in the discussion section of the manuscript which addresses the question of timing in more detail. For your and the reviewer's convenience, we have copy-pasted this paragraph here which we have now included in the discussion section of the manuscript (page 28 and the following):

“Another important aspect that will require further investigation in the future is the timing of the observed effects, particularly concerning the onset and progression of phenotypic changes in immune cell populations (i.e. Tregs, microglia and macrophages). While the activation of macrophages and microglia in response to hypercaloric feeding can be seen as early as one week after exposure to the HFHS diet, the reduction of Tregs in the hypothalamus occurs more gradually. Specifically, we identify a modest but significant reduction in Tregs in the hypothalamus after eight weeks of HFHS diet. Despite the observed decline in hypothalamic Tregs in response to HFHS diet, the assessment of any functional impairments in local Tregs that are caused by this hypercaloric challenge remains currently experimentally unfeasible, since only very few cells can be isolated from such hypothalami of mice. Due to these experimental limitations it is currently not possible to draw precise direct conclusions on whether Tregs in the hypothalamus exhibit a different functional profile/polarization/signature upon exposure to the HFHS diet and prior to their decline in the hypothalamus.“

In the following paragraphs, we would like to outline our actions taken following the specific sub-comments made by reviewer 4 in response to the points raised by reviewer 3. These actions have the goal to most appropriately integrate and accommodate both reviewers 3 and 4.

- o GOF experiments only tell what CAN happen, not what actually does happen

Response: We would like to indicate that reviewer 3 acknowledged that the initially performed experiments suggest a causal role of Tregs. As a compromise (since here reviewer 4 seems not to agree with reviewer 3), in this carefully revised version of the manuscript we have now rephrased the corresponding paragraphs to take the comments of reviewer 4 into account. For further details, please refer to **pages 21-24** of the manuscript, covering the gain-of-function experiments and especially page 24 where we have included a summary sentence:

“Taken together, these series of gain-of-function experiments suggest that Tregs in the CNS have the potential to lower diet-induced immune activation and to improve systemic metabolism.“

o Upregulation of *Tnfrsf14* is a picked cherry of unknown significance

Response: We acknowledge that the upregulation of *Tnfrsf14* on hypothalamic T cells from HFHSD-exposed is one of several genes that were differentially regulated. In the point-by-point reply, *Tnfrsf14* was rather mentioned as a representative example but by far not the only one mentioned in the manuscript. In contrast, in the results section of the manuscript we have detailed additional examples and cover a lot more information on *Adam10*, *Rock1*, *Irak1*, *Irf1*, *Mtor*, (*Tnfrsf14*) and *Il7r*. These paragraphs can be found on page 16.

o Treg suppression assays on lymphoid organ Tregs are not meaningful in this context.

Response: Additionally, to guide the reader and in line with the comment made by reviewer 4, we have now included a short paragraph in the manuscript, detailing why experiments addressing direct readouts of Treg function (such as *in vitro* suppression assays) are experimentally not feasible using Tregs from hypothalami of mice given the low numbers that can be isolated for such functional assays. In line with this, as recommended by reviewer 3, we have expanded the discussion further to provide a more detailed rationale for why suppression assays were performed using lymph node-derived Tregs (**Figure S1j**). The results obtained here indicate that the diet *per se* had no direct effects on the general Treg population. This addresses the question raised by reviewer 3 “whether the diet is affecting Treg function prior to their loss from the hypothalamus”.

The corresponding paragraph is included on page 11 and in the discussion.

For your convenience, these paragraphs are copy-pasted from the respective sections of the manuscript here:

(page 11) [...] Given the limited numbers of Tregs within the hypothalamus, it is not feasible experimentally to directly assess their cellular functionality through Treg suppression assays. However, we have performed such Treg suppression assays using Tregs isolated from LNs of either mice exposed to HFHS diet or to standard diet. These functional studies did not reveal any significant impairments in Treg function *per se* in mice exposed to the HFHS diet (**Figure S1j**). Nevertheless, our data indicate that the deteriorating effects of hypercaloric feeding on Tregs is a local effect as observed specifically in the hypothalamus.

(discussion) [...] Another important aspect that will require further investigation in the future is the timing of the observed effects, particularly concerning the onset and progression of phenotypic changes in immune cell populations (i.e. Tregs, microglia and macrophages). While the activation of macrophages and microglia in response to hypercaloric feeding can be seen as early as one week after exposure to the HFHS diet, the reduction of Tregs in the hypothalamus occurs more gradually. Specifically, we identify a modest but significant reduction in Tregs in the hypothalamus after eight weeks of HFHS diet. Despite the observed decline in hypothalamic Tregs in response to HFHS diet, the assessment of any functional impairments in local Tregs that are caused by this hypercaloric challenge remains currently experimentally unfeasible, since only very few cells can be isolated from such hypothalami of

mice. Due to these experimental limitations, it is currently not possible to draw precise direct conclusions on whether Tregs in the hypothalamus exhibit a different functional profile/polarization/signature upon exposure to the HFHS diet and prior to their decline in the hypothalamus.

To address the question whether the hypercaloric environment affects the whole Treg population in general, we used analyses from peripheral lymph-node derived Tregs. These analyses from peripheral lymph nodes showed that a HFHS diet *per se* does not have a major impact on Treg functionality. A direct assessment of hypothalamic Treg functionality is experimentally not possible currently, as highlighted above. Therefore, we used a series of independent Treg depletion strategies followed by exposure of animals to a hypercaloric challenge in order to dissect the functional relevance of Tregs in this anatomical niche. [...]

• Concerning “frequent” vs “frequency”: the reviewer is correct and the authors are incorrect. If others have used “frequent” in this context, they must be non-native English speakers.

Response: This was adapted accordingly.

References:

1. Feuerer M, *et al.* Lean, but not obese, fat is enriched for a unique population of regulatory T cells that affect metabolic parameters. *Nat Med* **15**, 930-939 (2009).
2. Li C, *et al.* TCR Transgenic Mice Reveal Stepwise, Multi-site Acquisition of the Distinctive Fat-Treg Phenotype. *Cell* **174**, 285-299 e212 (2018).
3. Becker M, *et al.* Regulatory T cells require IL6 receptor alpha signaling to control skeletal muscle function and regeneration. *Cell Metab* **35**, 1736-1751 e1737 (2023).
4. Ito M, *et al.* Brain regulatory T cells suppress astrogliosis and potentiate neurological recovery. *Nature* **565**, 246-250 (2019).

Point-by-point reply

Manuscript NCOMMS-22-21119B

REVIEWERS' COMMENTS

Reviewer #4 (Remarks to the Author):

This revised version should be accepted for publication.

We would like to thank Reviewer #4 for the positive feedback and recommendation to publish the manuscript.